# Non-stationary extreme value analysis applied to seismic fragility assessment for nuclear safety analysis

Jeremy Rohmer[1], Pierre Gehl[1], Marine Marcilhac-Fradin[2], Yves Guigueno[2], Nadia Rahni[2], Julien Clément[2]

[1]BRGM, 3 av. C. Guillemin, 45060 Orléans Cedex 2, France
[2]Institute for Radiological Protection and Nuclear Safety, Fontenay-Aux-Roses, 92262, France

*Correspondence to*: Jeremy Rohmer (j.rohmer@brgm.fr)

**Abstract.** Fragility curves (FC) are key tools for seismic probabilistic safety assessments that are performed at the level of the nuclear power plant (NPP). These statistical methods relate the probabilistic seismic hazard loading at the given site and the required performance of the NPP safety functions. In the present study, we investigate how the tools of non-stationary extreme value analysis can be used to model in a flexible manner the tail behaviour of the engineering demand parameter as a function of the considered intensity measure. We focus the analysis on the dynamic response of an anchored steam line and of a supporting structure under seismic solicitations. The failure criterion is linked to the exceedance of the maximum equivalent stress at a given location of the steam line. A series of three-component ground-motion records (~300) were applied at the base of the model to perform non-linear time history analyses. The set of numerical results was then used to derive a FC, which relates the failure probability to the variation of peak ground acceleration (*PGA*). The probabilistic model of the FC is selected via information criteria completed by diagnostics on the residuals, which support the choice of the generalized extreme value GEV distribution (instead of the widely used log-normal model). The GEV distribution is here non-stationary and the relationships of the GEV parameters (location, scale and shape) are established with respect to *PGA* using smooth non-linear models. The procedure is data-driven, which avoids the introduction of any a priori assumption on the shape/form of these relationships. To account for the uncertainties in the mechanical and geometrical parameters of the structures (elastic stiffness, damping, pipeline thicknesses, etc.), the FC is further constructed by integrating these uncertain parameters. A penalisation procedure is proposed to set to zero the variables of little influence in the smooth non-linear models. This enables us to outline which of these parametric uncertainties have negligible influence on the failure probability as well as the nature of the influence (linear, non-linear, decreasing, increasing, etc.) with respect to each of the GEV parameters.

## 1 Introduction

A crucial step of any seismic Probability Risk Assessment (PRA) is the vulnerability analysis of structures, systems and components (SSC) with respect to the external loading induced by earthquakes. To this end, fragility curves (FC), which relate the probability of an SSC to exceed a predefined damage state as a function of an intensity measure (*IM*) representing the hazard loading, are common tools. Formally, FC expresses the conditional probability with respect to the *IM* value (denoted

*im*) and to the *EDP* engineering demand parameter obtained from the structural analysis (e.g. force, displacement, drift ratio, etc.) as follows:

$$P_f(im) = P(EDP \geq th | IM = im), \tag{1}$$

where *th* is an acceptable demand threshold.

FCs are applied on a large variety of different structures like residential buildings (e.g. Gehl et al., 2013), nuclear power plant (Zentner et al., 2017), wind turbines (Quilligan et al., 2012), underground structures (Argyroudis and Pitilakis, 2012), etc. Their probabilistic nature make them well suited for PRA applications, at the interface between probabilistic hazard assessments and event tree analyses, in order to estimate the occurrence rate of undesirable top events.

Different procedures exist to derive FCs (see e.g. an overview by Zentner et al., 2017). In the present study, we focus on the analytical approach, which aims at deriving a parametric cumulative distribution function (CDF) from data collected from numerical structural analyses. A common assumption in the literature is that the logarithm of *im* is normally distributed (e.g., Ellingwood, 2001) as follows:

$$P_f(im) = \Phi\left(\frac{\log(im) - \log(\alpha)}{\beta}\right), \tag{2}$$

where $\Phi$ is the standard normal cumulative distribution function, $\alpha$ is the median and $\beta$ is lognormal standard deviation. The parameters of the normal distribution are commonly estimated either by maximum likelihood estimation (see e.g., Shinozuka et al., 2000) or by fitting a linear probabilistic seismic demand model in the log-scale (e.g., Banerjee and Shinozuka, 2008). This procedure faces, however, limits in practice:

- *Limit (1)*: the assumption of normality may not always be valid in all situations as discussed by Mai et al. (2017) and Zentner et al. (2017). This widely-used assumption is especially difficult to justify when the considered EDP corresponds to the maximum value of the variable of interest (for instance maximum transient stress value), i.e. when the FC serves to model extreme values;

- *Limit (2)*: a second commonly-used assumption is the homoscedasticity of the underlying probabilistic model, i.e. the variance term $\beta$ is generally assumed to be constant over the domain of the *IM*;

- *Limit (3)*: the assumption of linearity regarding the relation between the median and *IM* may not always hold valid as shown for instance by Wang et al. (2018) using artificial neural networks;

- *Limit (4)*: a large number of factors may affect the estimate of $P_f$ in addition to *IM*; for instance epistemic uncertainties due to the identification/characterization of some mechanical (e.g., elastic stiffness, damping ratio, etc.) and geometrical parameters of the considered structure.

The current study aims at going a step forward in the development of seismic FCs by improving the procedure regarding the afore-mentioned limits. To deal with limit (1), we propose to rely on the tools of extreme value statistics (Coles, 2001) and more specifically on the Generalised Extreme Value (GEV) distribution, which can model different extremes' behaviour.

Note that the focus is on the extremes related to EDP, not on the forcing, i.e. the analysis does not model the extremes of *IM* as it is done for current practices of probabilistic seismic hazard analysis (see e.g., Dutfoy, 2019). This means that no preliminary screening is applied, which implies that the FC derivation is conducted by considering both large and intermediate earthquakes, i.e. small-to-moderate to large *IM* values.

The use of GEV is examined using criteria for model selection like Akaike or Bayesian Information Criteria (Akaike, 1998;
Schwarz, 1978). Limits (2) and (3) are addressed using tools for distributional regression (e.g., Koenker et al., 2013) within the general framework of Generalized Additive Model for Location, Scale and Shape parameter (GAMLSS; e.g., Rigby and Stasinopoulos, 2005). GAMLSS is very flexible in the sense that the mathematical relation of the median and variance in Eq. 1 can be learnt from the data via nonlinear smooth functions. GAMLSS can be applied to any parametric probabilistic model, and here to the GEV model as a particular case. This enables us to fit a non-stationary GEV model, i.e. a GEV model for which
the parameters vary as a function of some covariates (here corresponding to *IM* and $\boldsymbol{U}$). The use of data-driven nonlinear smooth functions avoids introducing a priori model like linear or polynomial as many authors do (see an example by for sea level extremes by Wong (2018), and for temperature by Cheng et al., 2014).

Finally, accounting for the epistemic uncertainties in the FC derivation (limit (4)) can be conducted in different manners. A first option can rely on the incremental dynamic analysis (IDA), where the uncertain mechanical/geometrical parameters result
in uncertain capacities (i.e. related to the threshold *th* in Eq. (1)). The FC is then derived through convolution with the probabilistic distribution of the demand parameter; see Vamvatsikos and Cornell (2002). Depending on the complexity of the system (here for NPP), the adaptation of IDA to non-linear dynamic structural numerical simulations can be tedious (this is further discussed in Sect. 3.1). In the present study, we preferably opt for a second approach by viewing $P_f$ as conditional on the vector of uncertain mechanical and geometrical factors $\boldsymbol{U}$ (in addition to *IM*), namely:

$$P_f(im, \boldsymbol{u}) = P(EDP \geq th | IM = im, \boldsymbol{U} = \boldsymbol{u}), \tag{3}$$

Dealing with Eq. (3) then raises the question of integrating a potentially large number of variables, which might hamper the stability and quality of the procedure for FC construction. This is handled with a penalisation procedure (Marra and Wood,
2011), which enables the analysist to screen the uncertainties of negligible influence.

The paper is organised as follows. Section 2 describes the statistical methods to derive non-stationary GEV-based seismic fragility curves. Then, in Section 3, we describe a test-case related to the seismic fragility assessment for a steam line of a nuclear power plant. For this case, the derivation of FC is performed by considering the widely-used *IM* in the domain of seismic engineering, namely Peak Ground Acceleration (*PGA*). Finally, the proposed procedure is applied in Section 4 on two
cases, without and with epistemic uncertainties, and the results are discussed in Section 5.

## 2 Statistical methods

In this section, we first describe the main steps of the proposed procedure for deriving the FC (Sect. 2.1). The subsequent sections provide technical details on the GEV probability model (Sect. 2.2), its non-stationary formulation and implementation (Sect. 2.3) within the GAMLSS framework and its combination with variable selection (Sect. 2.4).

### 2.1 Overall procedure

To derive the seismic FC, the following overall procedure is proposed:

- Step 1 consists in analysing the validity of using the GEV distribution with respect to alternative probabilistic models (like the normal distribution of Eq. 2 in particular);

- Depending on the results of step 1, step 2 aims at fitting the non-stationary GEV model using the double penalisation formulation described in Sect. 2.2 and 2.3;

- Step 3 aims at producing some diagnostic information about the fitting procedure and results. The first diagnostic test uses the QQ plot of the model deviance residuals (conditional on the fitted model coefficients and scale parameter) formulated by Augustin et al. (2012). If the model distributional assumptions are met then the QQ plot should be close to a straight line. The second diagnostic test relies on a transformation of the data to a Gumbel distributed random variable (e.g. Beirlant et al., 2004) and on an analysis of the corresponding Gumbel QQ and PP plot;

- Step 4 aims at analysing the partial effect of each input variable (i.e. the smooth non-linear term, see Eq. 4 in Sect. 2.3) to assess the influence of the different GEV parameters;

- Step 5 aims at deriving the seismic FC by evaluating the failure probability $P_f(im, \boldsymbol{u}) = P(EDP \geq th | IM = im, \boldsymbol{U} = \boldsymbol{u})$. The following procedure is conducted to account for the epistemic uncertainties:

  - Step 5.1: the considered $IM$ is fixed at a given value;

  - Step 5.2: for the considered $IM$ value, a large number (here chosen at $n$=1,000) of $\boldsymbol{U}$ samples are randomly generated;

  - Step 5.3: for each of the randomly generated $\boldsymbol{U}$ samples, the failure probability is estimated for the considered $IM$ value;

  - Return to step 5.1.

The result of the procedure corresponds to a set of $n$ FCs from which we can derive the median FC as well as the uncertainty bands based on the pointwise confidence intervals at different levels. These uncertainty bands thus reflect the impact of the epistemic uncertainty related to the mechanical/geometrical parameters. Due to the limited number of observations, the derived FC is associated to the uncertainty on the fitting of the probabilistic model (e.g., GEV or Gaussian) as well. To integrate this fitting uncertainty in the analysis, step 5 can be extended by randomly generating parameters of the considered probabilistic model at step 5.2 (by assuming that they follow a multivariate Gaussian distribution).

## 2.2 Model selection

Selecting the most appropriate probabilistic models is achieved by means of information criteria as recommended in the domain of non-stationary extreme value analysis (e.g., Kim et al., 2017; Salas and Obeysekera, 2014), and more particularly recommended for choosing among various fragility models (e.g. Lallemant et al., 2015); see also an application of these criteria in the domain of nuclear safety by Zentner (2017). We focus on two information criteria, namely Akaike and Bayesian Information Criteria (Akaike, 1998; Schwarz, 1978), respectively denoted AIC and BIC whose formulation holds as follows:

$$\begin{aligned} \text{AIC} &= 2\log(l) + 2\text{k} \\ \text{BIC} &= 2\log(l) + \text{klog(n)}' \end{aligned}$$

(4)

where $l(.)$ is the log-likelihood of the considered probability model, k is the number of parameters, n is the size of the dataset used to fit the probabilistic model.

Though both criteria share similarities in their formulation, they provide different perspectives on model selection:

- AIC-based model selection considers a model to be a probabilistic attempt to approach the "infinitely complex data-generating truth – but only approaching not representing" (Höge et al. 2018: Table 2). This means that AIC-based analysis aims at addressing which model will predict the best the next sample, i.e. it provides a measure of the predictive accuracy of the considered model (Aho et al., 2014: Table 2);

- The purpose of BIC-based analysis considers each model as a "probabilistic attempt to truly represent the infinitely complex data-generating truth" (Höge et al. 2018: Table 2) assuming that the true model exists and is among the candidate models. This perspective is different from the one of AIC and focuses on an approximation of the marginal probability of the data (here lEDP) given the model (Aho et al., 2014: Table 2), and gives insights on which model generated the data, i.e. it measures goodness of fit.

The advantage of testing both criteria is to account for both perspectives on model selection, predictive accuracy and goodness of fit, while enabling to penalize too complex models; BIC generally penalizing more strongly than does the AIC. Since the constructed models use penalisation for the smoothness, we use the formulation provided by Wood et al. (2016: Sect. 5) to account for the smoothing parameter uncertainty.

Yet, selecting the most appropriate model may not be straightforward in all situations when two model candidates present close AIC/BIC values. For instance, Burnham & Anderson (2004) suggests an AIC difference (relative to the minimum value) of at least 10 to support the ranking between model candidates with confidence. If this criterion is not met, we propose to complement the analysis by the likelihood ratio test LRT (e.g., Panagoulia et al., 2014: Sect. 2), which compares two hierarchically nested GEV formulations using $L=-2(l_0-l_1)$, where $l_0$ is the maximized log-likelihood of the simpler model $M_0$ and $l_1$ is the one of the more complex model $M_1$ (that presents $q$ additional parameters compared to $M_0$ and contains $M_0$ as a particular case). The criterion $L$ follows a chi-squared distribution with $q$ degrees of freedom, which allows deriving a p-value of the test.

## 2.3 Non-stationary GEV distribution

The cumulative distribution function (CDF) of the Generalized Extreme Value (GEV) probability model holds as follows:

$$P(EDP \leq edp) = \exp\left(-\left(1 + \xi\left(\frac{edp - \mu}{\sigma}\right)\right)^{-1/\xi}\right), \tag{5}$$

where *edp* is the variable of interest; $\mu$, $\sigma$ and $\xi$ are the GEV location, scale, and shape parameters, respectively. Depending on the value of the shape parameter, the GEV distribution presents an asymptotic horizontal behaviour for $\xi < 0$ (i.e. the asymptotically–bounded distribution, which corresponds to the Weibull distribution); unbounded when $\xi > 0$ (i.e. high probability of occurrence of great values can be reached, which corresponds to the Fréchet distribution); and intermediate in the case of $\xi = 0$ (Gumbel distribution).

Fig. 1a illustrates the behaviour of the GEV density distribution for $\mu=12.5$, $\sigma=0.25$ and different $\xi$ values: the higher $\xi$, the heavier the tail. Fig. 1b,c further illustrates how changes in the other parameters (respectively the location and the scale) affect the density distribution. The location primarily translates the whole density distribution, while the scale affects the tail and to a lesser extent (for the considered case) the mode.

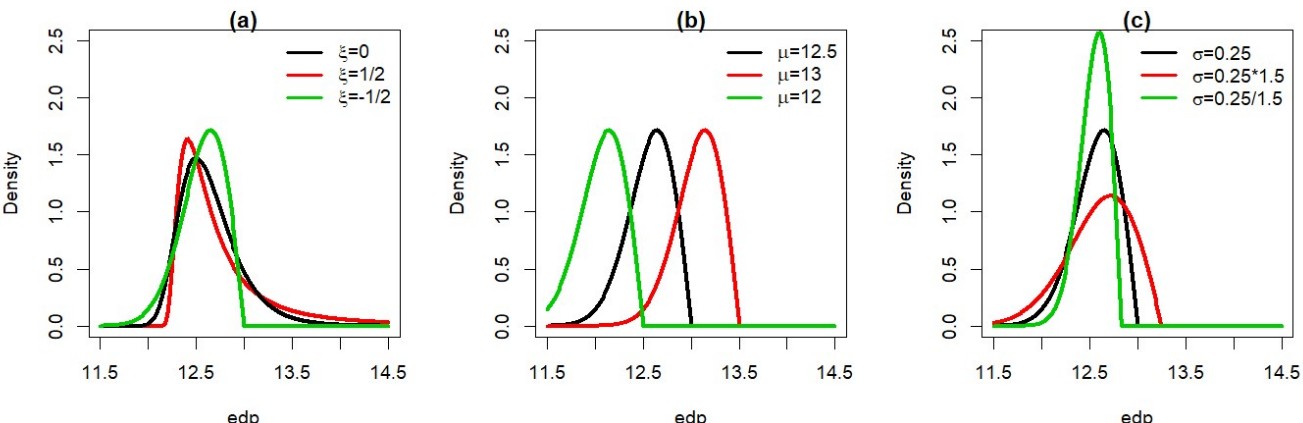

**Figure 1:** **Behaviour of the GEV density distributions depending on the changes in the parameter value: (a) $\xi$ (with $\mu$ fixed at 12.5, and $\sigma$ fixed at 0.25); (b) $\mu$ (with $\xi$ fixed at 0.5, and $\sigma$ fixed at 0.25); (c) $\sigma$ (with $\mu$ fixed at 12.5, and $\xi$ fixed at 0.5).**

The GEV distribution is assumed to be nonstationary in the sense that the GEV parameters $\boldsymbol{\theta}=(\mu,\sigma,\xi)$ vary as a function of **x** the vector of input variables, which include *IM* and the uncertain input variables **U** (as described in the introduction). The fitting is performed within the general framework of Generalized Additive Model for Location, Scale and Shape parameter (GAMLSS; e.g., Rigby and Stasinopoulos, 2005). Since the scale parameter satisfies $\sigma > 0$, we preferably work with its log-

transformation, which is denoted $l\sigma$. In the following, we assume that $\theta$ follows a semi-parametric additive formulation as follows:

$$\eta_\theta(x) = \sum_{j=1}^{J} f_j(x_j), \tag{6}$$

where $J$ is the number of functional terms that is generally inferior to the number of input variables (see Sect. 2.3), $f_j(.)$ corresponds to a univariate smooth non-linear model described as follows:

$$f_j(x) = \sum_b \beta_{jb} b_b(x), \tag{7}$$

with $b_b(.)$ the thin plate spline basis function (Wood, 2003) and $\beta_j$ the regression coefficients for the considered smooth function.

These functional terms (termed as partial effect) hold the information of each parameter's individual effect on the considered GEV parameter. The interest is to model the relationship between each GEV parameter and the input variables flexibly. Alternatives approach would assume a priori functional relationships (like linear or of polynomial form), which may not be valid.

The model estimation consists in evaluating the regression coefficients $\beta$ (associated to the GEV parameters $\theta$) by maximizing the log-likelihood $l(.)$ of the GEV distribution. To avoid overfitting, the estimation is based on the penalized version of $l(.)$ to control the roughness of the smooth functional terms (hence their complexity) as follows:

$$\underset{\beta}{\mathrm{argmax}}\left(l(\beta) - \tfrac{1}{2}\sum_j \lambda_j \beta^T S^j \beta\right), \tag{8}$$

where $\lambda_j$ controls the extent of the penalisation (i.e. the trade-off between goodness-of-fit and smoothness), and $S^j$ is a matrix of known coefficients (such that the terms in the summation measure the roughness of the smooth functions). Computational methods and implementation details are detailed in (Wood et al., 2016 and references therein). In particular, the penalisation parameter is selected through minimisation of the generalized cross validation score.

**2.4 Variable selection**

The introduction of the penalisation coefficients in Eq. 8 has two effects: they can penalize how "wiggly" a given term is (i.e. it has a smoothing effect) and they can penalize the absolute size of the function (i.e. it has a shrinkage effect). The second effect is of high interest to screen out input variables of negligible influence. However, the penalty can only affect the components that have derivatives, i.e. the set of smooth non-linear functions termed as the "range space". Completely smooth functions (including constant or linear functions), which belong to the "null space" are however not influenced by Eq. 8. For instance, for one-dimensional thin plate regression splines, a linear term might be left in the model, even when the penalty value is very large (as $\lambda \to \infty$); this means that the afore-described procedure does not ensure that an input variable of negligible

influence will completely be filtered out of the analysis (with corresponding regression coefficient shrunk to zero). The consequence is that Eq. 6 does not usually remove a smooth term from the model altogether (Marra and Wood, 2011). To overcome this problem, a double-penalty procedure was proposed by Marra and Wood (2011) based on the idea that the space of a spline basis can be decomposed in the sum of two components, one associated with the functions in the penalty null space and the other with the penalty range space. See Appendix A for further implementation details. This double-penalty procedure is adopted in the following.

To exemplify how the procedure works, we apply it on the following synthetic case. Consider a non-stationary GEV distribution whose parameters are related to two covariates $x_1$ and $x_2$ (see Eq. 9) as follows:

$$
\begin{aligned}
f_\mu(x) &= x_1{}^3 + 2.\,x_2{}^2 + 1 \\
f_{l\sigma}(x) &= x_1{}^2 \qquad , \\
f_\xi(x) &= -0.1
\end{aligned}
\tag{9}
$$

A total of 200 random samples are generated by drawing $x_1$ and $x_2$ from a uniform distribution on [0; 4] and [0; 2] respectively. Fig. 2a provides the partial effects for the synthetic test case using the single penalisation approach. The non-linear relationships are clearly identified for $\mu$ (Fig. 2a-i,ii) and for $l\sigma$ (Fig. 2a-ii). Yet, the single penalisation approach fails to identify properly the absence of influence of $x_2$ on $l\sigma$ and of both covariates on $\xi$ (Fig. 2a-iv,v,vi) since the resulting partial effects still present a linear trend (though with small amplitude and large uncertainty bands). Fig. 2b provides the partial effects using the double penalisation approach. Clearly, this type penalisation achieves a more satisfactory identification of the negligible influence of $x_2$ on $l\sigma$ and of both covariates on $\xi$ (Fig. 2b-iv,v,vi) as well the nonlinear partial effects for $\mu$ (Fig. 2b-i,ii) and for $l\sigma$ (Fig. 2b-ii).

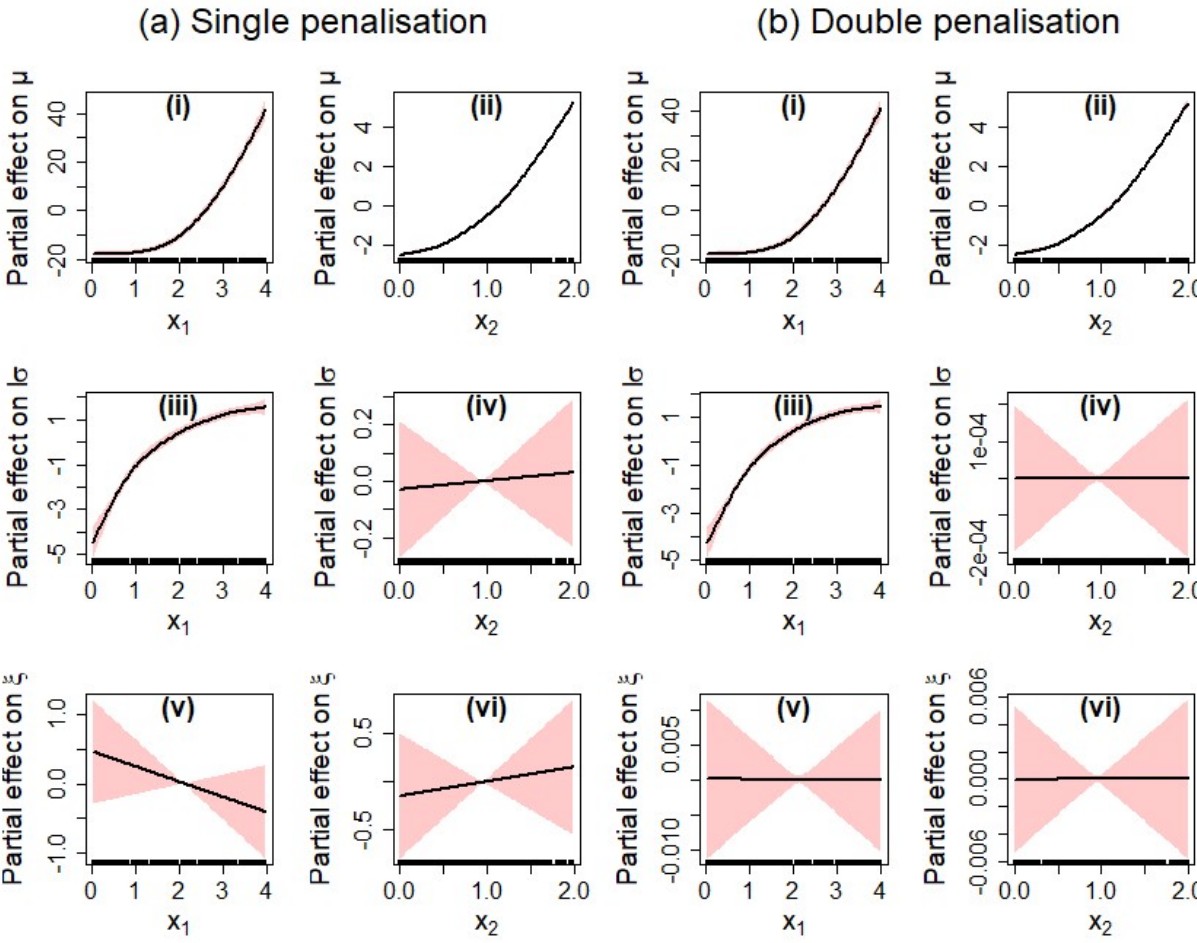

**Figure 2: Partial effect for the synthetic test case using the single penalisation approach (a) and the double penalisation approach (b).**

## 3 Application case

This section provides details on the test-case on which the proposed statistical methods (Sect. 2) for the derivation of FCs are demonstrated. The numerical model of the main steam line of a nuclear reactor is described in Sect. 3.1. A set of ground-motion records (Sect. 3.2) is applied to assess the seismic fragility of this essential component of a nuclear power plant.

### 3.1 Structural model

The 3-D model of a steam line and its supporting structure (i.e., the containment building, see schematic overview in Fig. 3a), previously assembled by Rahni et al. (2017) in the CAST3M finite-element software (Combescure et al., 1982), is introduced

here as an application of the seismic fragility analysis of a complex engineered object. The containment building consists of a double-wall structure: the inner wall (reinforced pre-stressed concrete) and the outer wall (reinforced concrete) are modelled with multi-degree-of-freedom stick elements (see Fig. 3b). The steel steam line is modelled by means of beam elements, representing pipe segments and elbows, as well as several valves, supporting devices and stops at different elevations of the supporting structure.

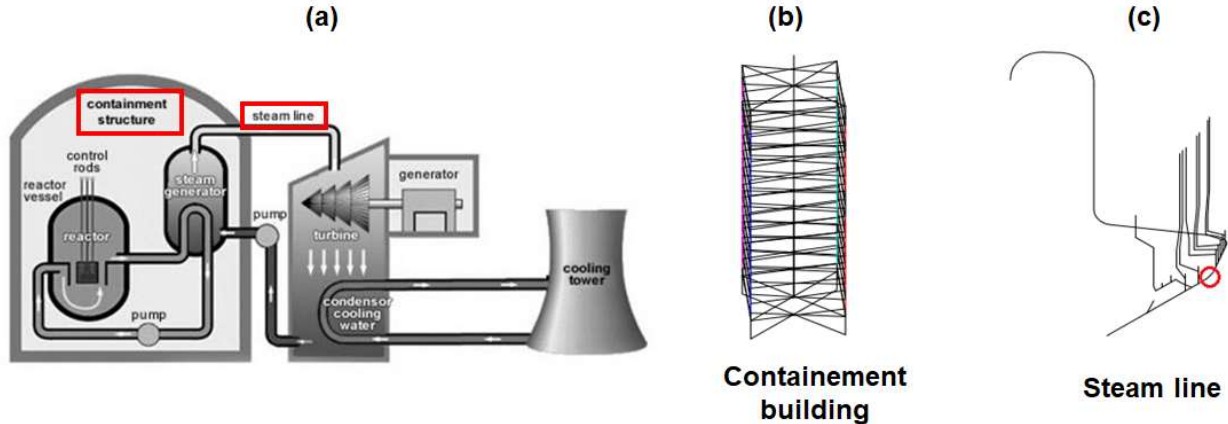

Figure 3: (a) Schematic overview of a nuclear power plant (adapted from nucleus.iaea.org); The red rectangles indicate the main components represented in the structural model. (b) Stick model of the containment building; (c) Steam line beam model, originally built by Rahni et al. (2017). The red circle indicates the location of the vertical stop.

The objective of the fragility analysis is to check the integrity of the steam line: one of the failure criteria identified by Rahni et al. (2017) is the effort calculated at the location corresponding to a vertical stop along the steam line (Fig. 3c). Failure is assumed when the maximum transient effort exceeds the stop's design effort, i.e. $EDP \geq 775$ kN (i.e. 13.56 in log-scale). The model also accounts for epistemic uncertainties due to the identification of some mechanical and geometrical parameters; namely the Young's modulus of the inner containment, the damping ratio of the structural walls and of the steam line, and the thickness of the steam line along various segments of the assembly. The variation range of the ten selected parameters, constituting the vector $U$ of uncertain factors (see Eq. 3), is detailed in Table 1. A uniform distribution is assumed for these parameters following the values provided by Rahni et al. (2017).

Table 1. Input parameters of the numerical model, according to Rahni et al. (2017).

| Variable | Description | Uniform distribution interval |
|---|---|---|
| $E_{IC}$ | Young's Modulus – Inner containment | [27700 – 45556] MPa |
| $\xi_{RPC}$ | Damping ratio – reinforced pre-stressed concrete | [4 – 6] % |
| $\xi_{RC}$ | Damping ratio – reinforced concrete | [6 – 8] % |

| $e_1$ | Pipe thickness – Segment #1 | [29.8 – 38.3] mm |
|---|---|---|
| $e_2$ | Pipe thickness – Segment #2 | [33.3 – 42.8] mm |
| $e_3$ | Pipe thickness – Segment #3 | [34.1 – 43.9] mm |
| $e_4$ | Pipe thickness – Segment #4 | [33.3 – 42.8] mm |
| $e_5$ | Pipe thickness – Segment #5 | [53.4 – 68.6] mm |
| $e_6$ | Pipe thickness – Segment #6 | [34.1 – 43.9] mm |
| $\xi_{SL}$ | Damping ratio – steam line | [1 – 4] % |

## 3.2 Dynamic structural analyses

A series of non-linear time-history analyses are performed on the 3-D model by applying ground-motion records (i.e., acceleration time-histories) at the base of the containment building in the form of a 3-component loading. In the CAST3M software, the response of the building is first computed, and the resulting displacement time-history along the structure is then applied to the steam line model, in order to record the effort demands during the seismic loading. The non-linear dynamic analyses are performed on a high performance-computing cluster, enabling the launch of the multiple runs in parallel (e.g., a ground-motion of a duration of 20s is processed in around 3 or 4 hours). Here, the main limit with respect to the number of ground-motion records is not necessarily related to the computation time cost, but more to the availability of natural ground motions that are able to fit the conditional spectra at the desired return periods (as detailed below). Another option would be the generation of synthetic ground motions, using for instance the stochastic simulation method by Boore (2003) or the non-stationary stochastic procedure by Pousse et al. (2006). It has been decided however to use only natural records in the present application, in order to accurately represent the inherent variability of other ground motion parameters such as duration.

Natural ground-motion records are selected and scaled using the conditional spectrum method described by Lin et al. (2013). Thanks to the consideration of reference earthquake scenarios at various return periods, the scaling of a set of natural records is carried out to some extent, while preserving the consistency of the associated response spectra. The steps of this procedure hold as follows:

- *Choice of a conditioning period*: the spectral acceleration (SA) at $T^* = 0.38$s (fundamental mode of the structure) is selected as the ground-motion parameter upon which the records are conditioned and scaled.

- *Definition of seismic hazard levels*: six hazard levels are arbitrarily defined, and the associated annual probabilities of exceedance are quantified with the OpenQuake engine[1], using the SHARE seismic source catalogue (Woessner et al., 2013), for an arbitrary site in Southern Europe. The GMPE from Boore et al. (2014) is used to generate the ground

---

[1] www.globalquakemodel.org

motions, assuming soil conditions corresponding to $V_{s,30}$ = 800 m/s at the considered site. Data associated with the mean hazard curve are summarized in Table 2.

Table 2: Estimation of the seismic hazard distribution for the application site.

| Scaling level | SA(0.38s) [g] | Annual Probability of Exceedance | Return Period |
|---|---|---|---|
| #1 | 0.185 | 4.87E-2 | 20 y |
| #2 | 0.617 | 4.99E-3 | 200 y |
| #3 | 0.836 | 2.50E-3 | 400 y |
| #4 | 1.492 | 5.00E-4 | 2,000 y |
| #5 | 2.673 | 5.00E-5 | 20,000 y |
| #6 | 3.882 | 5.00E-6 | 200,000 y |

- *Disaggregation of the seismic sources and identification of the reference earthquakes*: the OpenQuake engine is used to perform a hazard disaggregation for each scaling level. A reference earthquake scenario may then be characterized through the variables [$M_w$; $R_{jb}$; $\varepsilon$] (i.e., magnitude, Joyner-Boore distance, error term of the ground-motion prediction equation), which are averaged from the disaggregation results (Bazzurro and Cornell, 1999). This disaggregation leads to the definition of a mean reference earthquake (MRE) for each scaling level.

- *Construction of the conditional spectra*: for each scaling level, the conditional mean spectrum is built by applying the GMPE to the identified MRE. For each period $T_i$, it is defined as follows (Lin et al., 2013):

$$\mu_{ln\,SA(T_i)|\,ln\,SA(T^*)} = \mu_{ln\,SA}\left(M_w, R_{jb}, T_i\right) + \rho_{T_i,T^*} \cdot \varepsilon(T^*) \cdot \sigma_{ln\,SA}(M_w, T_i) \tag{10}$$

where $\mu_{lnSA}(M_w, R_{jb}, T_i)$ is the mean output of the GMPE for the MRE considered, $\rho_{T_i,T^*}$ is the cross-correlation coefficient between $SA(T_i)$ and $SA(T^*)$ (Baker and Jayaram, 2008), $\varepsilon(T^*)$ is the error term value at the target period $T^* = 0.38$s, and $\sigma_{lnSA}(M_w, T_i)$ is the standard deviation of the logarithm of $SA(T_i)$, as provided by the GMPE. The associated standard deviation is also evaluated, thanks to the following equation:

$$\mu_{ln\,SA(T_i)|\,ln\,SA(T^*)} = \mu_{ln\,SA}\left(M_w, R_{jb}, T_i\right) + \rho_{T_i,T^*} \cdot \varepsilon(T^*) \cdot \sigma_{ln\,SA}(M_w, T_i) \tag{11}$$

The conditional mean spectrum and its associated standard deviation are finally assembled in order to construct the conditional spectrum at each scaling level. The conditional mean spectra may be compared with the uniform hazard spectra (UHS) that are estimated from the hazard curves at various periods. As stated in Lin et al. (2013), the SA

value at the conditioning period corresponds to the UHS, which acts as an upper-bound envelope for the conditional mean spectrum.

- *Selection and scaling of the ground-motion records*: ground-motion records that are compatible with the target conditional response spectrum are selected, using the algorithm by Jayaram et al. (2011): the distribution of the selected ground-motion spectra, once scaled with respect to the conditioning period, has to fit the median and standard deviation of the conditional spectrum that is built from Eq. 10 and 11. The final selection from the PEER database (PEER, 2013) consists of 30 records for each of the 6 scaling levels (i.e., 180 ground-motion records in total).

Two distinct cases are considered for the derivation of FCs, depending on whether parametric uncertainties are included in the statistical model or not:

- *Case #1* (*without epistemic uncertainties*): A first series of numerical simulations are performed by keeping the mechanical and geometrical parameters fixed at their best estimate values, i.e. the mid-point of the distribution intervals detailed in Table 1. The 180 ground-motion records are applied to the deterministic structural model, resulting in a database of 180 *IM-EDP* points, with *PGA* chosen as the *IM*.
- *Case #2* (*with epistemic uncertainties*): A second series of numerical simulations are performed by accounting for parametric uncertainties. This is achieved by randomly varying the values of the mechanical and geometrical input parameters of the numerical model (Table 1) using the Latin Hypercube Sampling technique (Mc Kay et al., 1979). A total number of 360 numerical simulations are performed (using 180 ground-motion records).

Therefore, multiple ground motions are scaled at the same *IM* value, and statistics on the exceedance rate of a given *EDP* value may be extracted at each *IM* step, in a similar way as what is carried out in multi-stripe analyses or incremental dynamic analyses (Baker, 2015; Vamvatsikos and Cornell, 2002) for the derivation of FC. In the present study, the conditional spectrum method leads to the selection and scaling of ground motions with respect to $SA(0.38s)$, which corresponds to the fundamental modal of the structure. For illustration purposes, Fig. 4 displays the damage probabilities at the 6 selected return periods, which may be associated to unique values of $SA(0.38s)$.

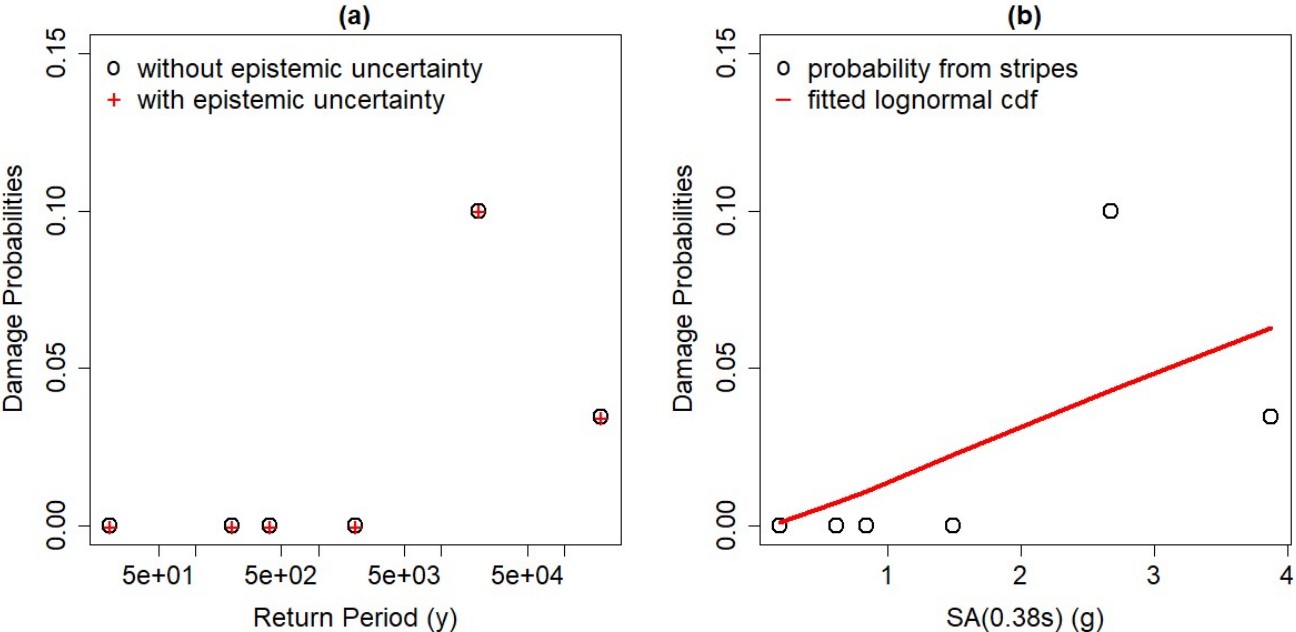

**Figure 4.** (a) Damage probabilities directly extracted from the 6 scaling levels (or return periods); (b) Damage probabilities w.r.t. the 6 SA(T*) levels, and fitted lognormal cumulative distribution function.

From Fig. 4, two main observations can be made: (i) the multiple stripe analysis does not emphasize any different between the models with and without parametric uncertainty, and (ii) the FC directly derived from the 6 probabilities does not provide a satisfying fit. However, the fragility analysis is here focused on the pipeline component (located along the structure), which appears to be more susceptible to *PGA*: therefore, *PGA* is chosen as *IM* in the present fragility analysis.

    Fig. 5 provides the evolution of l*EDP* (log-transformed of *EDP*) versus l*PGA* (log-transformed *PGA*) for both cases. We can

note that only a few simulation runs (5 for Case #1 and 8 for Case #2) lead to the exceedance of the acceptable demand threshold. As shown in Fig. 4, there is a variability around the 6 scaling levels: for this reason, it is not feasible to represent probabilities at 6 levels of *PGA*. In this case, conventional approaches for FC derivation are the 'regression on the *IM-EDP* cloud' (i.e., least-squares regression, as demonstrated by Cornell et al., 2002) or the use of Generalized Linear Model regression or maximum likelihood estimation (Shinozuka et al., 2000).

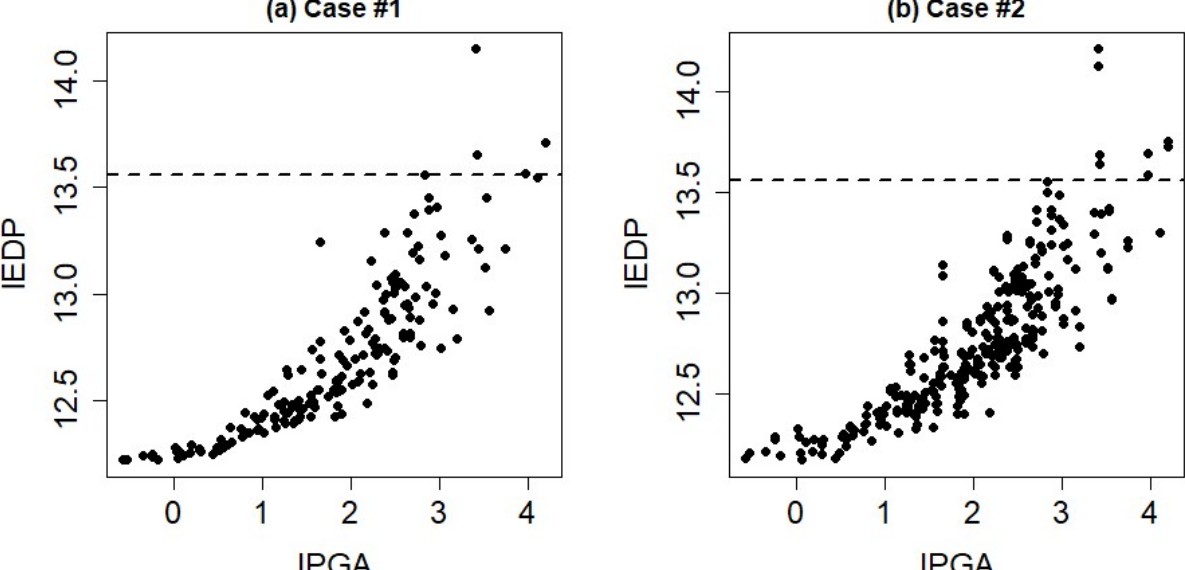

**Figure 5**: Evolution of l*EDP* (log-transformed *EDP*) as a function of l*PGA* (log-transformed *PGA*) for Case #1 (a) without parametric uncertainty, and for Case #2 (b) with parametric uncertainty. The horizontal dashed line indicates the acceptable demand threshold.

## 4 Applications

In this section, we apply the proposed procedure to both cases described in Sect. 3.2. Sect. 4.1 and 4.2 respectively describes the application for deriving the FCs without (case #1) and with epistemic uncertainty (case #2). For each case, we first select the most appropriate probabilistic model, then analyse the partial effects and finally, compare the derived FC with the one based on the commonly-used assumption of normality. The analysis is here focused on the log-transformed *PGA* (denoted l*PGA*) to derive the FC.

### 4.1 Case #1 Derivation of seismic FC without epistemic uncertainties

#### 4.1.1 Model selection and checking

A series of different probabilistic models (Table 3) were fitted to the database of *IM-EDP* points described in Sect. 3.2 (Fig. 5a). Three different probabilistic models (Normal, Tweedie, GEV) and two types of effects on the probabilistic model's parameters were tested (linear or non-linear). Note that the Tweedie distribution corresponds to a family of exponential distributions which takes as special cases the Poisson distribution and the Gamma distribution (Tweedie, 1984).

Table 3. Description of the probabilistic model used to derive the FCs

| Model name | Probability model | Type of relationship |
|---|---|---|
| NOsta | Normal | Stationary (without covariate effect) |

| NOlin1 | Normal | Linear effect on the mean |
|---|---|---|
| NOlin2 | Normal | Linear effect on the mean and log-transformed standard deviation (with link function log(σ+b), where b=0.01) |
| NOsmo1 | Normal | Non-Linear smooth effect on the mean |
| NOsmo2 | Normal | Non-Linear smooth effect on the mean and log-transformed standard deviation (with link function log(σ+b), where b=0.01) |
| GEVsta | GEV | Stationary (without covariate effect) |
| GEVlin1 | GEV | Linear effect on the location |
| GEVlin2 | GEV | Linear effect on the location and scale (log-transformed) |
| GEVlin3 | GEV | Linear effect on the location, scale (log-transformed) and shape |
| GEVsmo1 | GEV | Non-Linear smooth effect on the location |
| GEVsmo2 | GEV | Non-Linear smooth effect on the location and scale (log-transformed) |
| GEVsmo3 | GEV | Non-Linear smooth effect on the location, scale (log-transformed) and shape |
| TWElin1 | Tweedie | Linear effect on the log-transformed location |
| TWEsmo1 | Tweedie | Non-Linear smooth effect on the log-transformed location |

The analysis of the AIC/BIC differences (relative to the minimum value, Fig. 6) here suggests that both models, GEVsmo3 and GEVsmo2 are valid (as indicated by the AIC/BIC differences close to zero). The differences between the criteria value is less than 10, and to help the ranking, we complement the analysis by evaluating the LRT p-value, which reaches ~18%, hence suggesting that GEVsmo2 should be preferred (for illustration, the LRT p-value for a stationary GEV model and the non-stationary GEVsmot2 model is here far less than 1%). In addition, we also analyse the regression coefficients of GEVsmo3, which shows that the penalisation procedure imposes all coefficient of the shape parameters to be zero, which indicates that l$PGA$ only acts on the location and scale parameters.

These results provide support in favour of GEVsmo2, i.e. a GEV distribution with non-linear smooth term for the location and scale parameters only. The estimated shape parameter reaches here a constant value of 0.07 (+/-0.05), hence indicating a behaviour close to the Gumbel domain. This illustrates the flexibility of the proposed approach based on GEV, which

encompasses the Gumbel distribution as a particular case. We also note that the analysis of the AIC and BIC values would
have favoured the selection of NOsmo2 if the GEV model had not been taken into account, i.e. a heteroscedastic log-normal
FC.

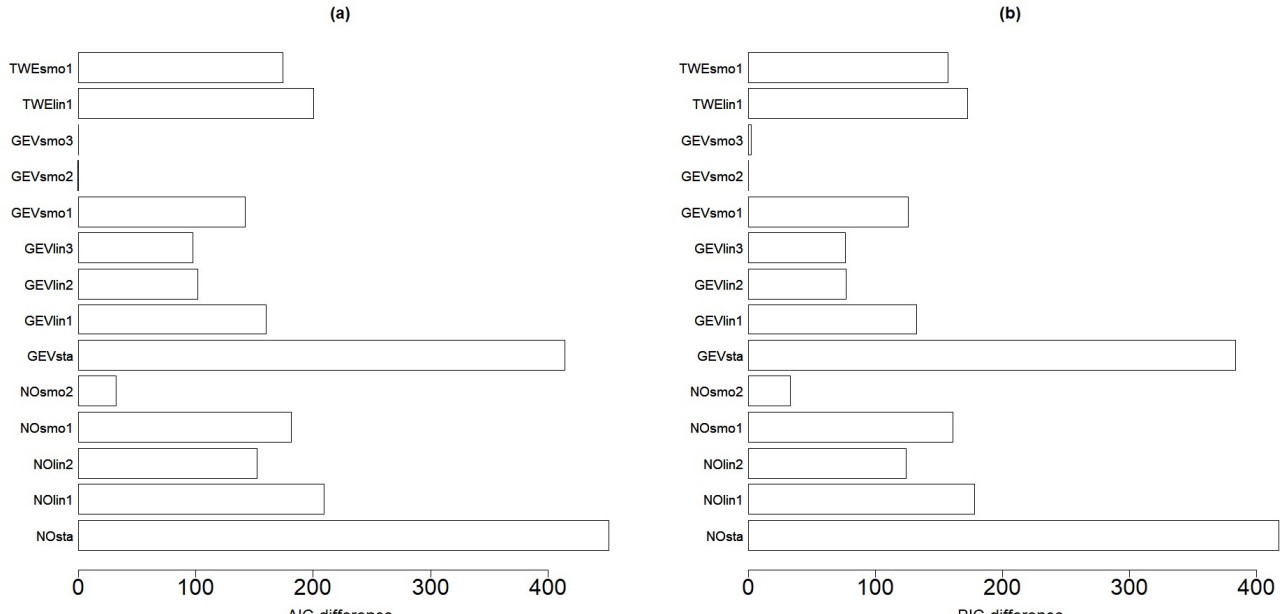

**Figure 6: Model selection criteria (AIC (a) and BIC (b) differences relative to the minimum value) for the different models described in Table 3 considering the derivation of a FC without epistemic uncertainty.**

The examination of the diagnostic plots (Fig. 7a) of the model deviance residuals (conditional on the fitted model coefficients and scale parameter) shows a clear improvement of the fitting; in particular for large theoretical quantiles above 1.5 (the dots better aligned along the first bisector in Fig. 7b). The Gumbel QQ and PP plots (Fig. 7c,d) also indicate a satisfactory fitting of the GEV model.

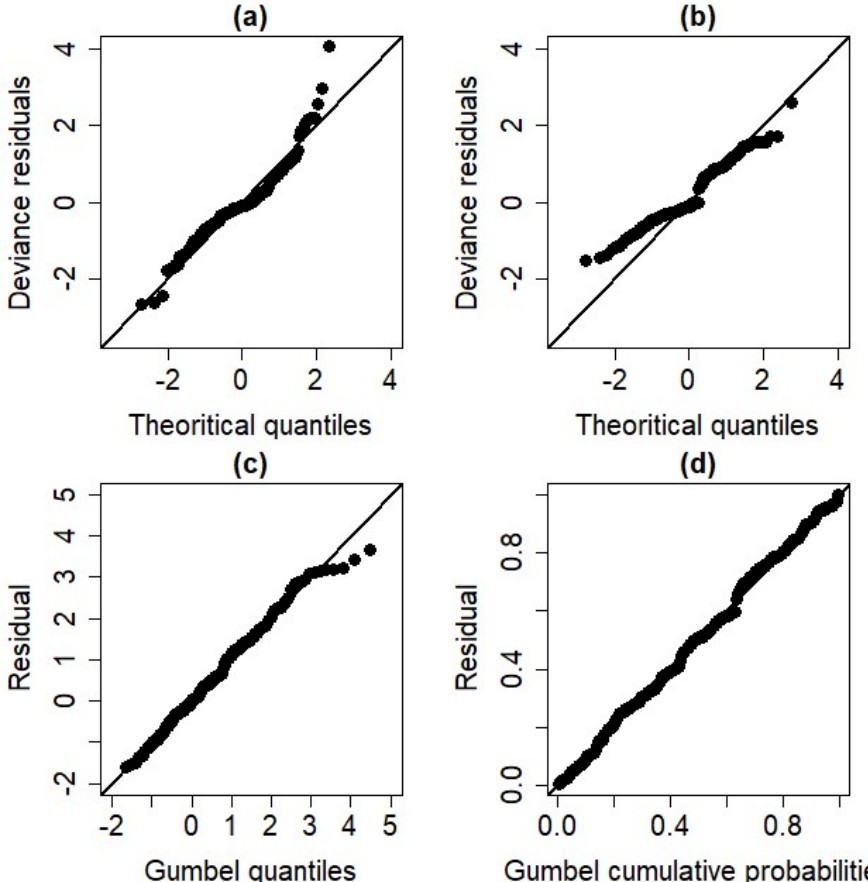

**Figure 7: Diagnostic plots to check the validity of the considered model: (a) QQ plot for the deviance residuals for the NOsmo2 model; (b) QQ plot for the deviance residuals for the GEVsmo2 model without parametric uncertainty; (c) QQ plot on Gumbel scale; (d) PP plot on Gumbel scale.**

### 4.1.2 Partial effects

Fig. 8a,b respectively provides the evolution of the partial effects (as formally described in Sect. 2.3: Eqs. 6 and 7) with respect to the location and to the log-transformed scale parameter. We note that the assumption of the relationship between *EDP* and

400 l*PGA* is non-linear (contrary to the widely-used assumption). An increase in l*PGA* both induces an increase of $\mu$ and of l$\sigma$, hence a shift of the density (as illustrated in Fig. 1b) and an impact on the tail (as illustrated in Fig. 1c). We note that the fitting uncertainty (indicated by the +/- two standard above and below the best estimate) remains small and the afore-described conclusions can be considered with confidence.

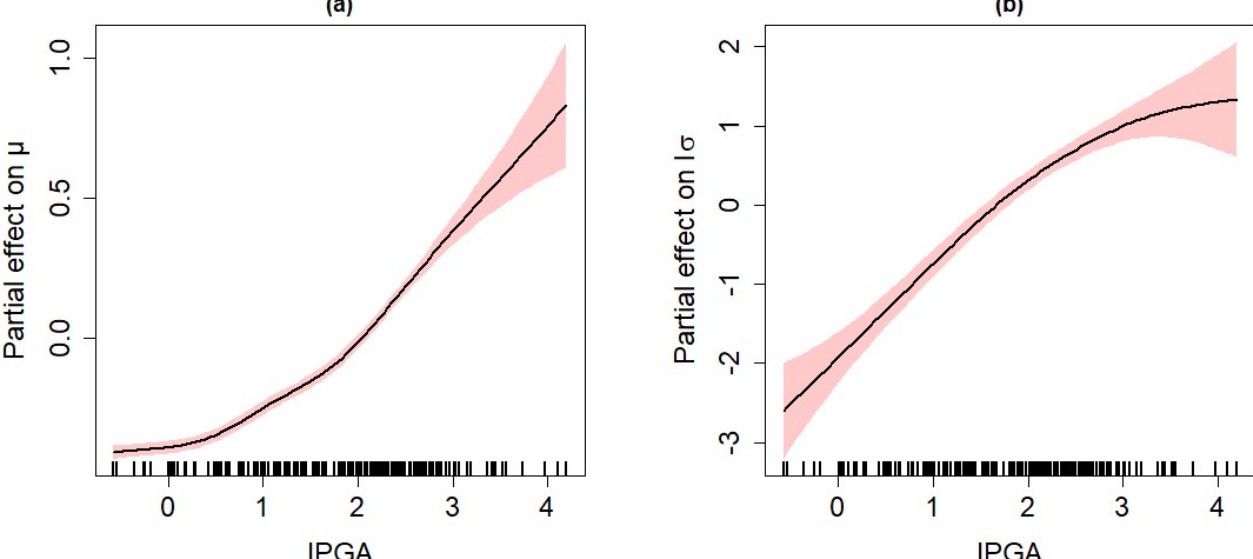

**Figure 8**: Partial effect of (a) PGA on the GEV location parameter; (b) PGA on the log-transformed GEV scale parameter. The red-coloured bands are defined by 2 standard errors above and below the estimate.

### 4.1.3 FC derivation

Using the Monte-Carlo-based procedure described in Sect. 2.1, we evaluate the failure probability $P_f$ (Eq. 1) to derive the corresponding GEV-based FC (Fig. 9a) with accounts for fitting uncertainties. The resulting FC is compared to the one based on the normal assumption (Fig. 9b). This shows that $P_f$ would have been under-estimated for moderate-to-large PGA from 10 to ~25 m²/s if the selection of the probability model had not been applied (i.e. if the widespread assumption of normality had been used); for instance at PGA=20 m²/s, $P_f$ is under-estimated by ~5%. This is particularly noticeable for the range of PGA from 10 to 15 m²/s, where the GEV-based FC clearly indicates a non-zero probability value, whereas the Gaussian model indicates negligible probability values below 1%. For very high PGA, both FC models approximately provide almost the same $P_f$ value. These conclusions should however be analysed with respect to the fitting uncertainty, which has here a clear impact; for instance at PGA=20 m²/s, the 90% confidence interval has a width of 10% (Fig. 9a), i.e. of the same order of magnitude than a PGA variation from 10 to 20 m²/s. We note also that the fitting uncertainty reaches the same magnitude between both models. This suggests that additional numerical simulation results are necessary to decrease this uncertainty source for both models.

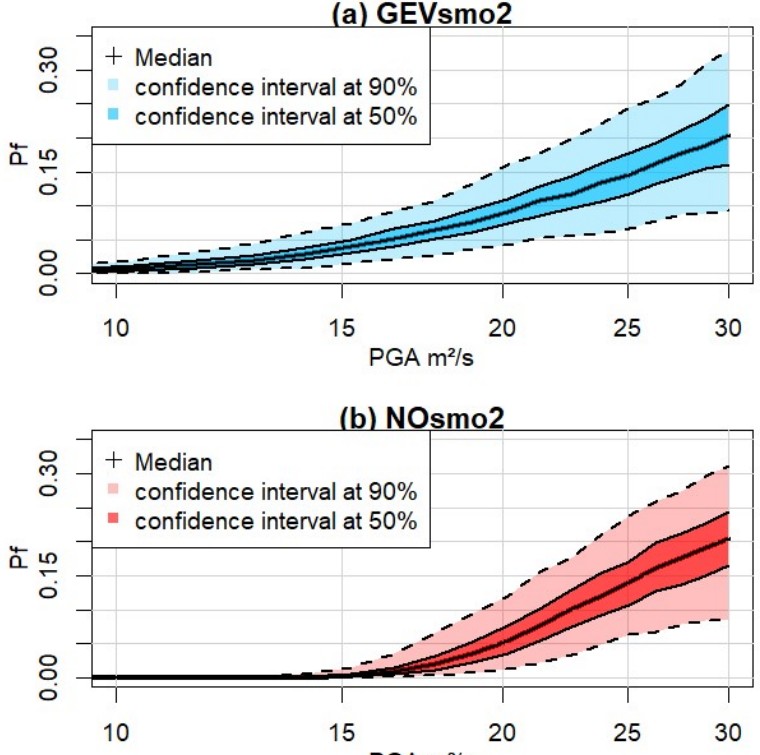

**Figure 9**: Fragility curve (relating the failure probability $P_f$ to *PGA*) based on (a) the non-stationary model GEVsmo2, (b) the Gaussian NOSmo2 model. The coloured bands reflect the uncertainty in the fitting.

## 4.2 Case #2 Derivation of seismic FC with epistemic uncertainties

### 4.2.1 Model selection and checking

In this case, the FCs were derived by accounting not only for *lPGA* but also for 10 additional uncertain parameters (Table 1).

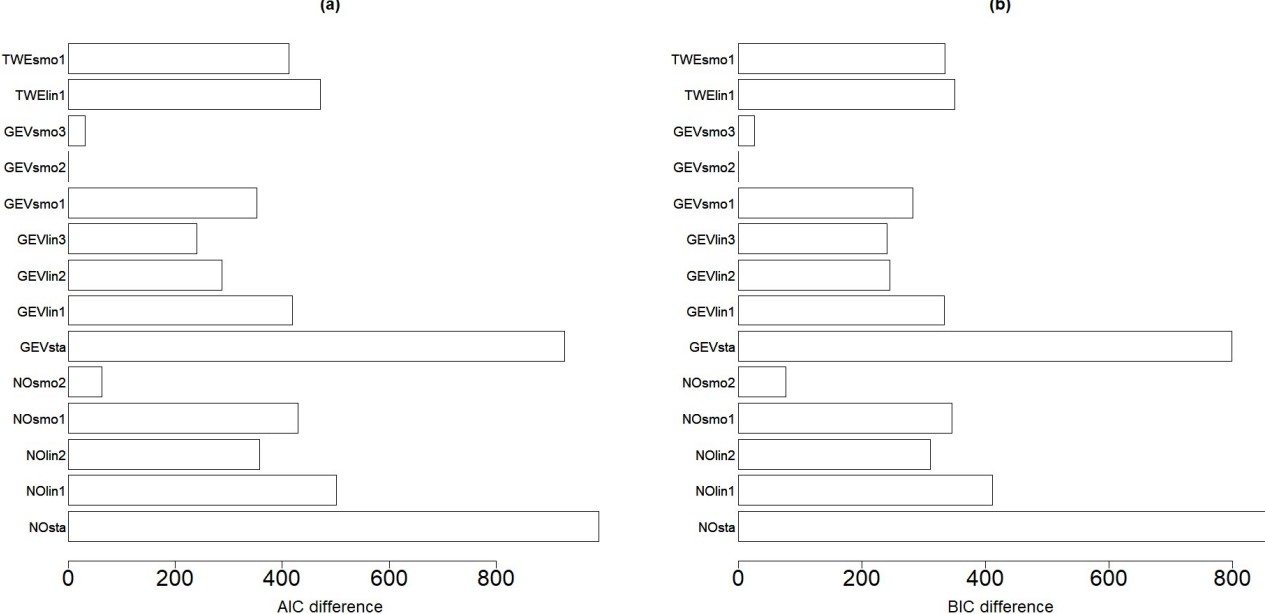

**Figure 10**: Model selection criteria (AIC (a) and BIC (b) differences relative to the minimum value) for the different models described in Table 3 considering the derivation of a FC with epistemic uncertainty.

The AIC and BIC differences ((relative to the minimum value, Fig. 10) for the different probabilistic models (described in Table 2) show that GEVsmo2 model should preferably be selected. Contrary to case #1, the AIC/BIC differences are large enough to rank with confidence GEVsmo2 as the most appropriate model. This indicates that the location and scale parameters are non-linear smooth functions of *IM* and of the uncertain parameters. The estimated shape parameter reaches here a constant value of -0.24 (+/-0.06), hence indicating a Weibull tail behaviour. Similarly as for the analysis without parametric uncertainties (Sect. 4.1), we note that the AIC and BIC values would have favoured the selection of NOsmo2 if the GEV model had not been taken into account.

The examination of the QQ plots (Fig. 11) of the model deviance residuals (conditional on the fitted model coefficients and scale parameter) shows an improvement of the fitting; in particular for large theoretical quantiles above 1.0 (the dots better aligned along the first bisector in Fig. 11b). The Gumbel QQ and PP plot (Fig. 11c,d) also indicate a very satisfactory fitting of the GEV model.

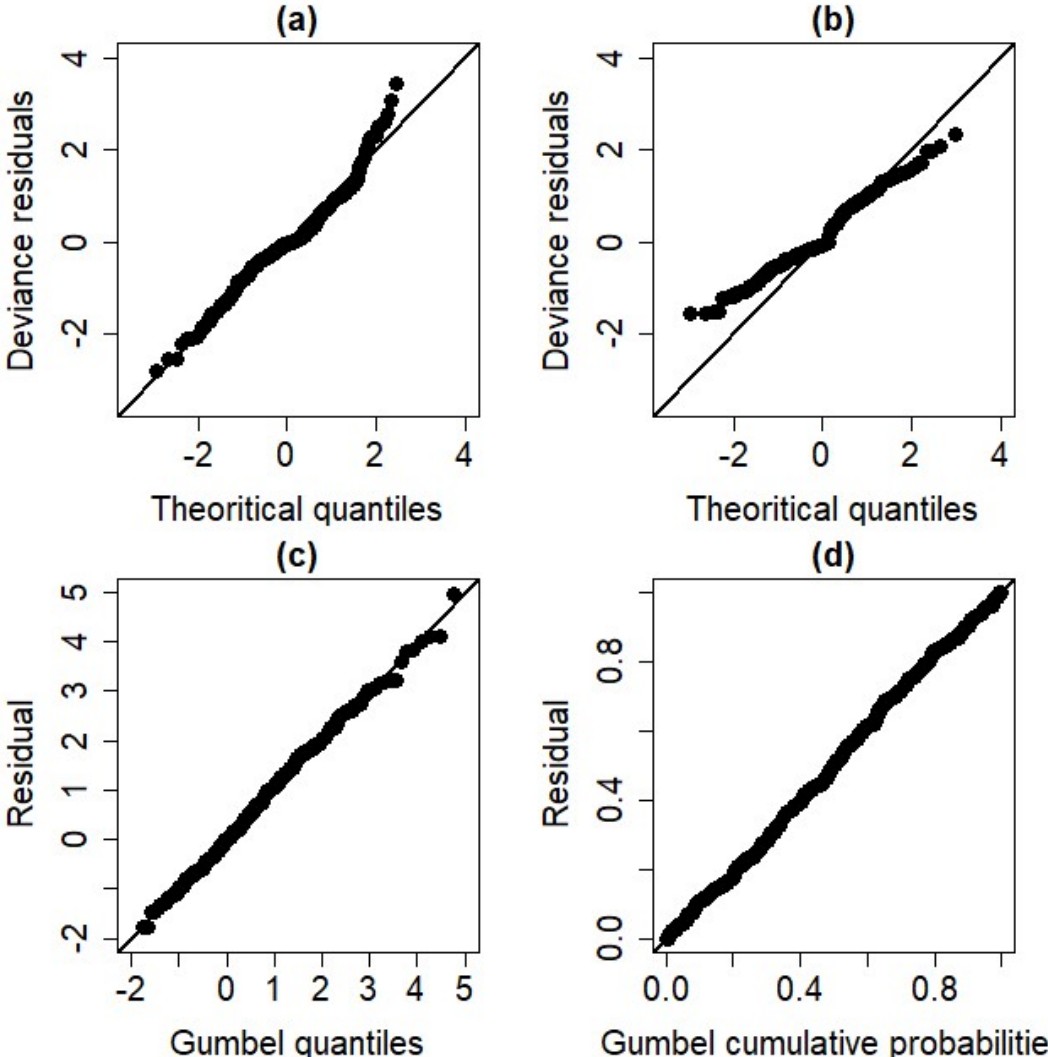

**Figure 11: Diagnostic plots to check the validity of the considered model: (a) QQ plot for the deviance residuals for the NOsmo2 model; (b) QQ plot for the deviance residuals for the GEVsmo2 model with epistemic uncertainty; (c) QQ plot on Gumbel scale; (d) PP plot on Gumbel scale.**

### 4.2.1 Partial effects

Fig. 12 provides the evolution of the partial effects with respect to the location parameter. Several observations can be made:

- Fig. 12a show quasi-similar partial effect for $lPGA$ (Fig. 8a);

- Three among the ten uncertain parameters were filtered out by the procedure of Sect. 2.4, namely two mechanical parameters (the damping ratio of reinforced pre-stressed concrete $\xi_{RPC}$, and the damping ratio of the steam line $\xi_{SL}$)

and one geometrical parameter (the pipe thickness of segment #2). As an illustration, Fig. 12e depicts the partial effect of a parameter, which was identified as of negligible influence: here, the partial effect of $e_2$ is shrunk to zero;

- Three thickness parameters ($e_1$, $e_4$, $e_5$) present an increasing linear effect on $\mu$ (Fig. 12d,g,h);
- Two parameters (the Young's Modulus of the inner containment $E_{IC}$ and the thickness $e_3$) present a decreasing linear effect on $\mu$ (Fig. 12b,f);
- The damping ratio of the reinforced concrete $\xi_{RC}$ presents a non-linear effect with a minimum value at around 0.0725 (Fig. 12c);
- The thickness $e_6$ presents a non-linear effect with a maximum value at around 0.04 (Fig. 12i).

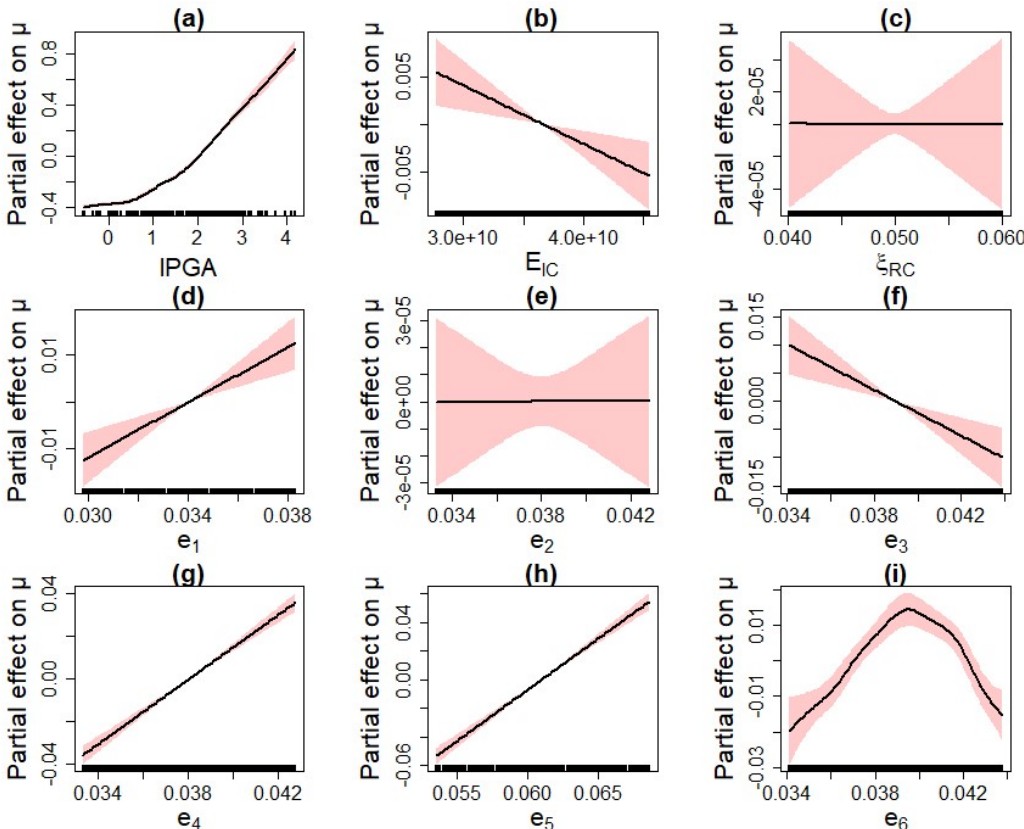

**Figure 12**: Partial effect on the GEV location parameter. The red-coloured bands are defined by 2 standard errors above and below the estimate.

Fig. 13 provides the evolution of the partial effects with respect to the (log transformed) scale parameter. We show here that a larger number of input parameters were filtered out by the selection procedure i.e. only the thickness $e_5$ is selected as well as the damping ratios of the concrete structures $\xi_{RPC}$ and $\xi_{RC}$ (related to the containment building). The partial effects are all non-

470 linear, but with larger uncertainty than for the location parameter (compare the widths of the red-coloured uncertain bands in Fig. 12 and 13). In particular, the strong non-linear influence of $\xi_{RPC}$ and $\xi_{RC}$ may be due to the simplified coupling assumption between structural dynamic response and anchored steam line (i.e., the displacement time-history at various points of the building is directly used as input for the response of the steam line). Identifying this problem is possible thanks to the analysis of the partial effects, though it should be recognized that this behavior remains difficult to interpret and further investigations

are here necessary. We also note that the partial effect for l$PGA$ is quasi-similar to Fig. 8b.

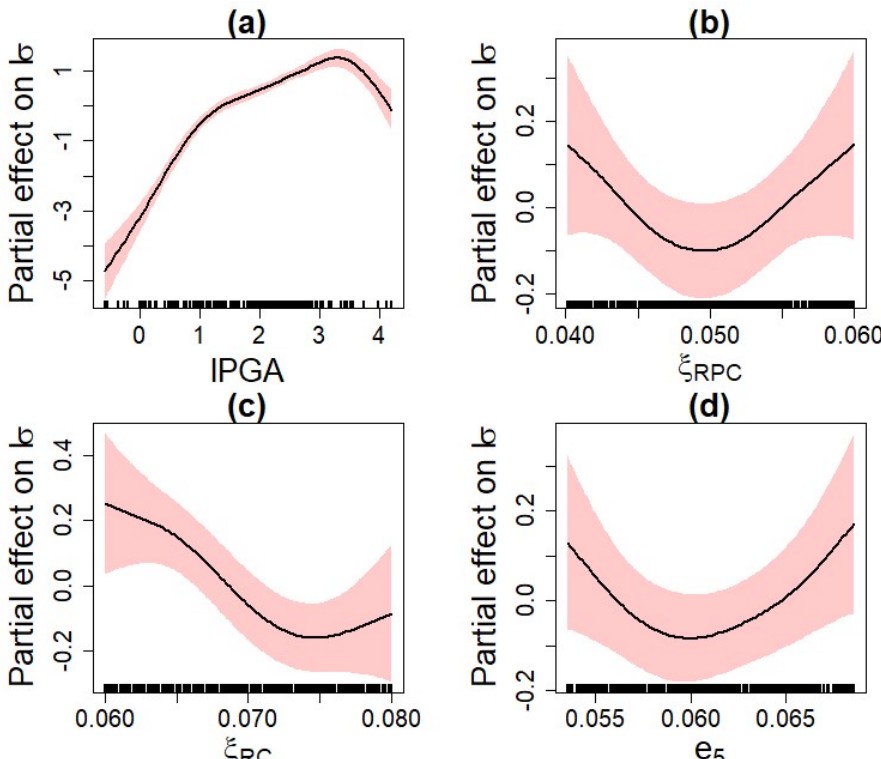

**Figure 13: Partial effect on the log-transformed GEV scale parameter. The red-coloured bands are defined by 2 standard errors above and below the estimate.**

Table 4 summarizes the different types of influence identified in Fig. 12 and 13, i.e. linear, non-linear, or absence of influence as well as the type of monotony when applicable.

Table 4. Influence of the geometrical/mechanical parameters on the GEV parameters, $\mu$ and l$\sigma$ of the GEVsmo2 model.

| Variable | Influence on $\mu$ | Influence on l$\sigma$ |
|----------|-------------------|------------------------|
| $E_{IC}$ | Linear (decreasing) | - |

| | | |
|---|---|---|
| $\xi_{RPC}$ | - | Non-linear (non-monotone) |
| $\xi_{RC}$ | Non-linear (non-monotone) | Non-linear (decreasing) |
| $e_1$ | Linear (increasing) | - |
| $e_2$ | - | - |
| $e_3$ | Linear (decreasing) | - |
| $e_4$ | Linear (increasing) | - |
| $e_5$ | Linear (increasing) | Non-linear (non-monotone) |
| $e_6$ | Non-linear (non-monotone) | - |
| $\xi_{SL}$ | - | - |

### 4.2.3 FC derivation

Based on the results of Fig. 12 and 13, the FC is derived by accounting for the epistemic uncertainties by following the Monte-Carlo procedure (step 5 described in Sect. 2.1) by including (or not) fitting uncertainty (Fig. 14a and b respectively). We show that the GEV-based FC is less steep than the one for case #1 (Fig. 9): this is mainly related to the value of the shape parameter

(close to Gumbel regime for case #1 without epistemic uncertainty and of Weibull regime for case #2 with epistemic uncertainty). Fig. 14a also outlines that the uncertainty related to the mechanical and geometrical parameters has a non-negligible influence as shown by the width of the uncertainty bands: for PGA=30 m²/s, the 90% confidence interval has a width of ~20%. In addition, the inclusion of the fitting uncertainty (Fig. 14b) increases the width of the confidence interval, but it appears to mainly impact the 90% confidence interval (compare the dark and the light coloured envelope in Fig. 14); for

instance, compared to Fig. 14a, this uncertainty implies a +5% (respectively -5%) shift of the upper bound (respectively lower bound) of the 90% confidence interval at PGA=30 m²/s.

Compared to the widely-used assumption of normality, Fig. 14c,d show that the failure probability reached with this model is larger than with the GEV-based FC; at PGA=30 m²/s, the difference reaches ~5%. In practice, this means that a design based on the Gaussian model would have here been too conservative. Regarding the impact of the different sources of uncertainty,

the epistemic uncertainty appears to influence less the Gaussian model than the GEV one. The impact of the fitting uncertainty is however quasi-equivalent for both models.

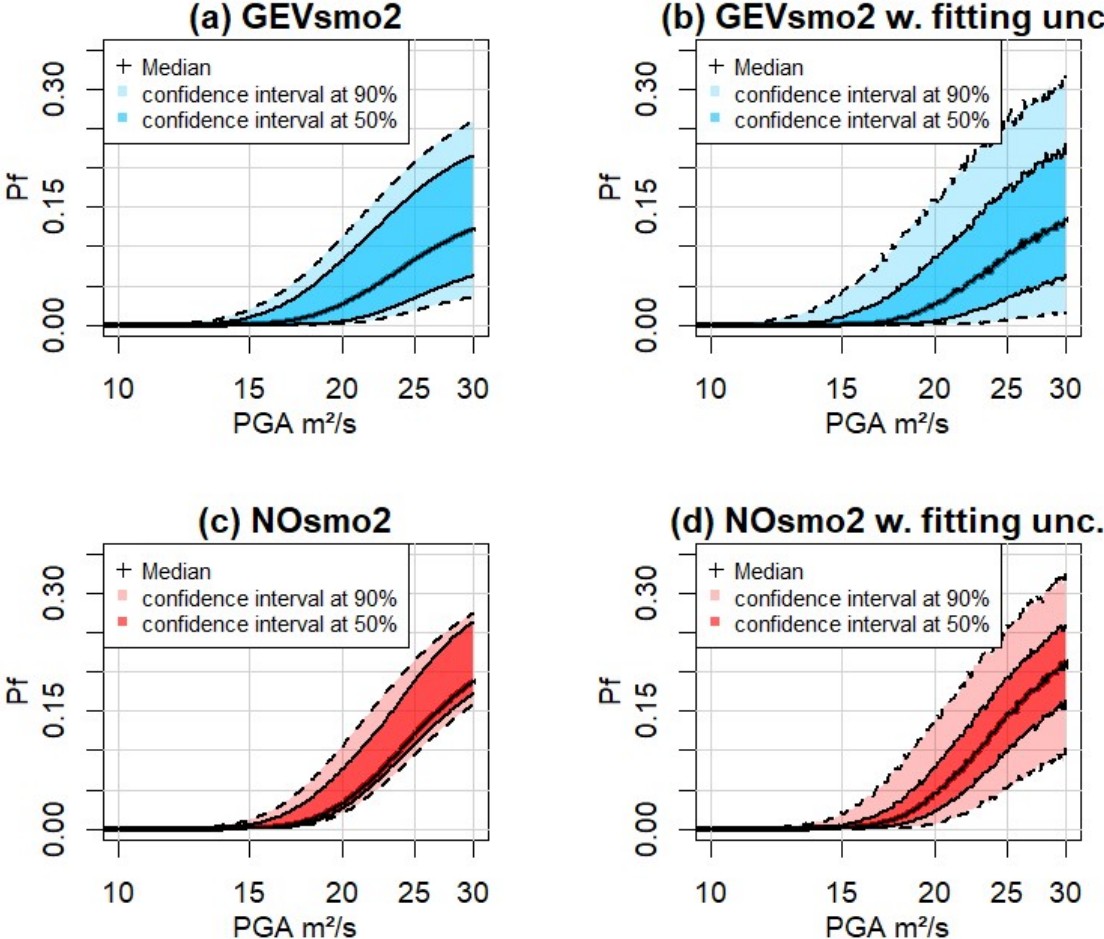

**Figure 14**: Fragility curve (relating the failure probability $P_f$ to *PGA*) considering epistemic uncertainties only (left), and fitting uncertainty as well (right). (a,b) GEV-based FC; (c,d) FC based on the normal assumption. The coloured bands are defined based on the pointwise confidence intervals derived from the set of FCs (see text for details).

The interest of incorporating the mechanical/geometrical parameters directly in the equation of the FC is the ability to study how the FC in Fig. 14 evolves as a function of the parametric uncertainties, hence to identify regions of the parameters' values leading to large failure probability. This is illustrated in Fig. 15, where the FC is modified depending on the value of the thickness $e_4$, from -12.5% (0.033m) to +12.5% (~0.043m) with respect to the median value of 0.038m. Here larger $e_4$ induces a steeper FC. This appears to be in agreement with the increasing effect of $e_4$ as shown in Fig. 12g. Fig. 15 also shows that the effect of $e_4$ on $P_f$ only becomes significant when the $e_4$ variation is of a least +/-5%, compared to the fitting uncertainty (of the order of magnitude of +/- 2.5%).

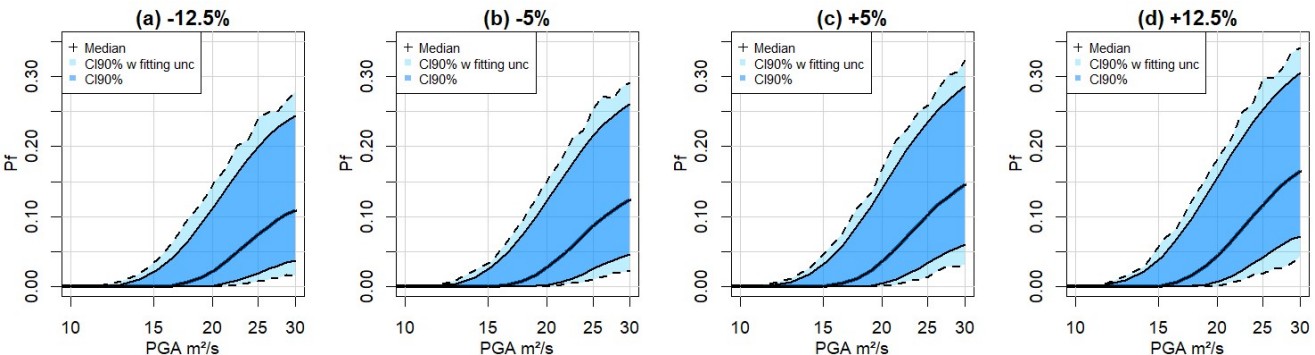

**Figure 15**: FC considering different thickness e4: (a) -12.5% of the original value; (b) -5%; (c) +5%; (d) +12.5%. Uncertainty bands are provided by accounting for epistemic uncertainty only (dark blue) and by accounting for the fitting uncertainty as well (light blue).

## 5 Discussion and further work

The current study has focused on the problem of seismic FC derivation for nuclear power plant safety analysis. We propose a procedure based on the non-stationary GEV distribution to model, in a flexible manner, the tail behaviour of the *EDP* as a function of the considered *IM*. The key ingredient is the use of non-linear smooth functional *EDP-IM* relationships (partial
effects) that are learnt from the data (to overcome limits (2) and (3) as highlighted in the introduction). This avoids the introduction of any a priori assumption on the shape/form of these relationships. In particular, the benefit is shown on case #1 (without epistemic uncertainty), where the non-linear relation is clearly outlined for both $\mu$ and l$\sigma$. The interest of such data-driven non-parametric techniques has also been shown using alternatives techniques (like neural network, Wang et al., 2018 or kernel smoothing, Mai et al., 2017). To bring these approaches to an operative level, an extensive comparison/benchmark
exercise on real cases should be conducted in the future.

The second objective of the present study was to compare the GEV-based FC with the one based on the Gaussian assumption. We show that if a careful selection of the most appropriate model is not performed (limit (1) described in the introduction), the failure probability would be either under- or over-estimated for case #1 (without epistemic uncertainty) and case #2 (with epistemic uncertainty), respectively. This result brings an additional element against the uncritical use of the (log-)normal
fragility curve (see discussions by Karamlou and Bocchini, 2015; Mai et al., 2017; Zentner et al., 2017, among others).

The third objective was to propose an approach to incorporate the mechanical and geometrical parameters in the FC derivation (using advanced penalisation procedures). The main motivation was to allow studying the evolution of the failure probability as function of the considered covariate (as illustrated in Fig. 15). As indicated in the introduction, an alternative approach would rely on the principles of the IDA method; the advantage being to capture the variability of the structural capacity and to
540 get deeper insight into the structural behaviour. See an example for masonry buildings by Rota et al. (2010). Yet, the adaptation

of this technique would impose additional developments to properly characterise collapse through the numerical model (see discussion by Zentner et al., 2017: Sect. 2.5). Sect. 3.1 also points out the difficulty in applying this approach in our case. Combining the idea underlying IDA and our statistical procedure is worth investigating in the future.

The benefit of the proposed approach is to provide information on the sensitivity to the epistemic uncertainties by both identifying the parameters of negligible influence (via the double penalisation method), and by using the derived partial effects. The latter hold information on the magnitude and nature of the influence (linear, non-linear, decreasing, increasing, etc.) for each GEV parameters (to overcome limit (4)). Additional developments should however be performed to derive the same levels of information for the FC (and not only for the parameters of the probabilistic model). In this view, Fig. 15 provides a first basis that can be improved by: 1) analysing the role of each covariate from a physical viewpoint, as done for instance by Salas and Obeysekera (2014) to investigate the evolution of hydrological extremes over time (e.g. increasing, decreasing or abrupt shifts of hydrologic extremes). Some valuable lessons can also be drawn from this domain of application to define and communicate an evolving probability of failure (named return period in this domain); 2) deriving a global indicator of sensitivity via variance-based global sensitivity analysis (see e.g., Borgonovo et al., 2013). The latter approach opens promising perspectives to ease the fitting process by filtering out beforehand some negligible mechanical/geometrical parameters. It is also expected to improve the interpretability of the procedure by clarifying the respective role of the different sources of uncertainty i.e. related to the mechanical/geometrical parameters, but also to the fitting process, which appears to have a non-negligible impact in our study.

The treatment of this type of uncertainty can be improved on two aspects: 1) it is expected to decrease by fitting the FC with a larger number numerical simulation results. To relieve the computational burden (each numerical simulation has a computation time cost of several hours, see Sect. 3.2), replacing the mechanical simulator by surrogate models (like neural network, Wang et al., 2018 or using model order reduction strategy, Bamer et al., 2017) can be envisaged; 2) the modelling of such uncertainty can be done in a more flexible and realistic manner (compared to the Gaussian assumption made here) using Bayesian techniques within framework of GAMLSS (Umlauf et al., 2018).

Finally, from an earthquake engineering viewpoint, the proposed procedure has focused on a single *IM* (here *PGA*), but any other *IM*s could easily be incorporated, similarly as for the mechanical and geometrical parameters, to derive vector-based FC as done by Gehl et al. (2019) using the same structure. The proposed penalisation approach can be seen as a valuable option to solve a recurrent problem in this domain, namely the identification of most important *IM*s (see discussion by Gehl et al., 2013 and references therein).

**Author contributions**

JR designed the concept with input from PG. M M-F, YG, NR, JC designed the structural model and provided to PG for adaptation, and implementation on the described case. PG performed the dynamical analyses. JR undertook the statistical analyses and wrote the paper, with inputs from PG.

**Competing interests**

The authors declare that they have no conflict of interest.

**Code/Data availability**

Code are available upon request to the first author. Statistical analysis was performed using R package mgcv (available at https://cran.r-project.org/web/packages/mgcv/index.html). See Wood (2017) for an overview. Numerical simulations were performed with CAST3M simulator (Combescure et al., 1982).

**Acknowledgements**

This study has been carried out within the NARSIS project, which has received funding from the European Union's H2020-Euratom Programme under grant agreement N° 755439. We thank both reviewers for their constructive comments, which led to the improvement of the manuscript.

**Appendix A Double penalisation procedure**

This appendix gives further details on the double penalisation procedure used to select variable in the nonstationary GEV. Full details are described by Marra and Wood (2011).

Consider the smoothing penalty matrix $\boldsymbol{S}^j$ in Eq. 6 (associated to the $j$th smooth function in the semi-parametric additive formulation of Eq. 4). This matrix can be decomposed as

$$\boldsymbol{U}_j \boldsymbol{\Lambda}_j \boldsymbol{U}_j^T, \tag{A1}$$

where $\boldsymbol{U}_j$ is the eigenvector matrix associated with the $j$th smooth function, and $\boldsymbol{\Lambda}_j$ is the corresponding diagonal eigenvalue matrix. As explained in Sect. 2.4, the penalty as described in Eq. 6 can only affect the components that have derivatives, i.e. the set of smooth non-linear functions termed as the "range space". Completely smooth functions (including constant or linear functions), which belong to the "null space" are however not influenced. This problem implies that $\boldsymbol{\Lambda}_j$ contains zero eigenvalues, which makes the variable selection difficult for "nuisance" functions belonging to the null space, i.e. functions

with negligible influence on the variable of interest. The idea of Marra and Wood (2011) is to introduce an extra penalty term which penalizes only functions in the null space of the penalty to achieve a complete removal of the smooth component. Consider the decomposition (A1), an additional penalty can be defined as $\boldsymbol{S}_j^* = \boldsymbol{U}_j^* \boldsymbol{U}_j^{*T}$ where $\boldsymbol{U}_j^*$ is the matrix of eigenvectors corresponding to the zero eigenvalues of $\boldsymbol{\Lambda}_j$. In practice, the penalty in Eq.6 holds as follows:

$$\lambda_j \boldsymbol{\beta}^{\mathrm{T}} \boldsymbol{S}_j \boldsymbol{\beta} \;+\; \lambda_j^* \boldsymbol{\beta}^{\mathrm{T}} \boldsymbol{S}_j^* \boldsymbol{\beta}\;, \tag{A2}$$

where two penalization parameters $(\lambda_j, \lambda_j^*)$ are estimated; here by minimization of the generalized cross validation score.

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
