# Peer review of "Non-stationary extreme value analysis applied to seismic fragility assessment for nuclear safety analysis"

_Natural Hazards and Earth System Sciences, 2019_

## Referee Comment (RC1) · Anonymous Referee #1 · 13 Jan 2020

**1. General comments**

The authors present in this manuscript a methodology to derive low probabilities of failure for a nuclear plant, based on a simplified numerical model, by fitting a statistical distribution to the response. The paper propose several non-linear models to link the response to the different covariates and some model selection to derive the best estimation of failing probability, called here Fragility Curve.

The paper well expose the models used, however some of them could be better explained, and the results when the covariates uncertainties are taken into account are well presented. In comparison, the description of the construction of the database is

less clear to me, and as is would be difficult to reproduce.

The paper is well written, with relevant references and good quality figures. The methods used, if not the newest, have not been already used in the domain, as far as I know. The application is sensible and realistic. The problem addressed is worth being published.

2. Specific comments

2.1 Statistical methods

The description of variable selection method is rather crude and could be better explained. For example, the double-penalty procedure is not presented, and would better serve the paper than the description of the GEV distributions. Moreover, it could be interesting to compare the results with a dedicated variable selection algorithm such as boosting for example (e.g. with gamboostLSS package). As is, it is difficult to understand how the selection if done and in particular how variables are excluded from the figures 10 and 11.

I also have some concerns about the model selection : since here the authors are not interested in predicting new values, are AIC and BIC the best selection criteria to use ? In particular, for an explanatory model, the QQ plots can be a better tool and may leeds to different conclusion. For example, in the case of parametric uncertainty, I would go for the Gumbel model (figure 9). Could the authors precise why the use AIC and BIC in this case and how could they go further ?

2.2 Application case

The selection of the ground-motion records if not described precisely enough from my point of view, for example the scaling levels are not stated. E.g. return levels for with quantity ? The records are non-linear and non-stationary in time, so how the spectrum is computed and scaled ?

The computational time for running seems to be omitted, it might be interesting to give

an idea if a more important database could be generated.

2.3 Results

The models compared here do not include parametric models (polynomials, non-linear...) and the selected models are the non-linear smooth models. One question is related to the ability of this models to extrapolate beyond the range of variation of the training set ? It might be interesting to compare to classical parametric models (if any) of with some polynomials models to also investigate the extrapolation ability.

If my understanding is correct, the uncertainties in the estimation of the marginal effects are neglected in computing the fragility curves, that is the reason why the are no uncertainties on figure 7. However, in figures 12 and 13, uncertainties linked to the variability of the input variables are shown. As is, it difficult to know which source of uncertainties is the highest and a discussion on this point would add a great value to the paper.

3. Technical remarcks

- Both formula, figures and tables should be centered to be easier to read; - in figures 10 and 11, some variables seems to be evenly distributed and some other (e.g. E_IC, \Xi_RC, e5 in figure 10) seems to be random : it seems that all of them should be uniform of the range of variation stated in Table 1 ? - The link functions are not stated precisely in table 2;

---

## Referee Comment (RC2) · Anonymous Referee #2 · 24 Feb 2020

**1   General comments**

The paper proposes a method to derive improved and accurate fragility curves for nuclear plants subject to seismic activity by adopting non-stationary GEV for the engineering demand parameter. The capacity of the structure is simulated allowing for parametric uncertainties. The premise of the proposed method is novel and reasonable.

The analysis is rigorous and encompasses major requirements of uncertainty quantification. The results indicate that the usage of the method presented may provide better

vulnerability assessment in nuclear plant safety analysis by a significant margin.

The paper may benefit from improved clarity of presentation, particularly in the final computation of fragility curves (FC) and presentation of results.

**2 Specific comments: Methods**

The choice of whether to use GEV is done using AIC or BIC measures. The benefit to using a non-stationary GEV may be demonstrated by showing improved goodness of fit or other custom measures as applicable. Fig.5 and 9 may be locations to include such a comparison.

The derivation of the fragility after arriving at demand and capacity may be done more explicitly. The nonlinear structural analysis section describes a scaling range with 6 steps. One might expect a few data points on the FC rather than a continuous curve based on this. A fit may be done after this and superimposed on the same graph. The absence of this plot may be due to the usage of a different method. The GEV fit appears to be for the EDP but is also mentioned as the fit for FC. Clarity here may improve readability considerably.

The same ambiguity arises in the plots of partial effects (Figs. 6,10,11). The structural variables appear to have a partial effect on the demand parameter. This seems counter-intuitive conventionally. The GEV appears to be used not just for demand in that case, this may be better presented.

The convolution of the probability density of capacity around the pre-defined damage states and the 1-CDF of the demand on the structure by different levels of ground motion would produce points on the FC. The procedure is detailed in [1]. This may be used as a starting point to show how FC derivation is different here.

Return period for a non-stationary model requires transformation which may be of significance in some cases. See [2]. This may be of no effect considering the order of scaling, but it may be of use to include/discuss.

**3 Readability**

The paper may benefit from an appendix or a different section for detailed methods after describing the main results. The partial effects may be consolidated in one section, the fragility curves being in another. The variation of the fragility curves based on the choice for parameters such as e4 may be better presented in measures of percentage changes. The method used by Wood et al. (line 97) could've included with more detail for completeness.

The paper may benefit from a tabular presentation of results, especially the effect of structural parameters on FC as this may be of key significance for a practitioner. Figure quality may be improved. Consider using vector graphics. x-axis of Fig. 5 requires uniformity and units may be placed in brackets.

[1] Rota, M., A. Penna, and G. Magenes. "A methodology for deriving analytical fragility curves for masonry buildings based on stochastic nonlinear analyses." Engineering Structures 32.5 (2010): 1312-1323.

[2] Salas, Jose D., and Jayantha Obeysekera. "Revisiting the concepts of return period and risk for nonstationary hydrologic extreme events." Journal of Hydrologic Engineering 19.3 (2014): 554-568.

---

## Author Response (AR1)

**Replies to the reviewers' comments on "Non-stationary extreme value analysis applied to seismic fragility assessment for nuclear safety analysis". (nhess-2019-400)**

We would like to thank both referees for their constructive comments. We agree with most of the suggestions and, therefore, we have modified the manuscript to take on board their comments. In the following, we recall the reviews and we reply to each of the comments in turn (outlined by "**<Authors' reply>**").

**Please note that the line numbers of changes are indicated and correspond to the revised manuscript with marked changes.**

**Referee #1:**

**1. General comments**
*The authors present in this manuscript a methodology to derive low probabilities of failure for a nuclear plant, based on a simplified numerical model, by fitting a statistical distribution to the response. The paper propose several non-linear models to link the response to the different covariates and some model selection to derive the best estimation of failing probability, called here Fragility Curve.*

*The paper well expose the models used, however some of them could be better explained, and the results when the covariates uncertainties are taken into account are well presented. In comparison, the description of the construction of the database is less clear to me, and as is would be difficult to reproduce.*

*The paper is well written, with relevant references and good quality figures. The methods used, if not the newest, have not been already used in the domain, as far as I know. The application is sensible and realistic. The problem addressed is worth being published.*

**2. Specific comments**

**2.1 Statistical methods**
*The description of variable selection method is rather crude and could be better explained. For example, the double-penalty procedure is not presented, and would better serve the paper than the description of the GEV distributions.*
**<Authors' reply>** We agree with referee #1 and have elaborated on this aspect by providing a complementary appendix. Besides, Sect. 2.4 has been completed with an application on a synthetic test case as follows:
"To exemplify how the procedure works, we apply it on the following synthetic case. Consider a nonstationary GEV distribution whose parameters are influenced by two covariates $x_1$ and $x_2$ (see Eq. 6) as follows:

$$f_\mu(x) = x_1{}^3 + 2.x_2{}^2 + 1$$
$$f_{l\sigma}(x) = x_1{}^2 \qquad , \qquad\qquad\qquad (6)$$
$$f_\xi(x) = -0.1$$

A total of 200 random samples are generated by drawing $x_1$ and $x_2$ from a uniform distribution on [0; 4] and [0; 2] respectively. Fig. 2a provides the partial effects for the synthetic test case using the single penalisation approach. The non-linear relationships are clearly identified for $\mu$ (Fig. 2a-i,ii) and for $l\sigma$ (Fig. 2a-ii). Yet, the single penalisation approach fails to identify properly the absence of influence of $x_2$ on $l\sigma$ and of both covariates on $\xi$ (Fig. 2a-iv,v,vi) since the resulting partial effects still present a linear trend (though with large uncertainty bands). Fig. 2b provides the partial effects using the double penalisation approach. Clearly, the penalisation achieves to identify the absence of influence (Fig. 2b-iv,v,vi) as well as the nonlinear partial effects for $\mu$ (Fig. 2b-i,ii) and for $l\sigma$ (Fig. 2b-ii)".

[Figure]

**New Figure 2:** Partial effect for the synthetic test case using the single penalisation approach (a) and the double penalisation approach (b).

*Moreover, it could be interesting to compare the results with a dedicated variable selection algorithm such as boosting for example (e.g. with gamboostLSS package). As is, it is difficult to understand how the selection if done and in particular how variables are excluded from the figures 10 and 11.*
**<Authors' reply>** The current version of gamboostLSS package does not consider the GEV distribution and adding these new functionalities to this specific package is out of the scope of the current study. We however agree that mentioning alternative fitting (and variable selection) approaches should be added to the manuscript as future lines of research.

Besides, to bring additional elements to referee #1 (and out of curiosity), we applied the gamboostLSS procedure by randomly generating 200 observations from a nonstationary Gumbel distribution considering the following relationships:

$$f_\mu(x) = x_1{}^3 + 2. x_2{}^2 + 1$$
$$f_{l\sigma}(x) = x_1{}^2$$

The following figure provides the comparison between the partial effect for $l\sigma$ derived from
    (a) the double-penalisation-based fit as proposed in the present work;
    (b) the boosting-based fit (using a 5-fold cross validation procedure combined with the noncyclical algorithm by Thomas et al. (2018) for selecting the stopping boosting cut-off).

[Figure]

**Figure.** Partial effect for the synthetic Gumbel test case using the double penalisation approach (a) and the gamboostLSS approach (b).

Both approaches achieve to identify the negligible influence of $x_2$ on $l\sigma$, but the magnitude of the influence remains small-to-moderate for gamboostLSS and highly dependent on selection of the stopping boosting cut-off. On this aspect, the double-penalisation procedure appears to be more robust. This should however be confirmed in a more extensive benchmark exercise that could be a line for future research of the present work.

**Reference**
Thomas, J., Mayr, A., Bischl, B., Schmid, M., Smith, A., & Hofner, B. (2018). Gradient boosting for distributional regression: faster tuning and improved variable selection via noncyclical updates. Statistics and Computing, 28(3), 673-687.

*I also have some concerns about the model selection : since here the authors are not interested in predicting new values, are AIC and BIC the best selection criteria to use ? In particular, for an explanatory model, the QQ plots can be a better tool and may leeds to different conclusion. For example, in the case of parametric uncertainty, I would go for the Gumbel model (figure 9). Could the authors precise why the use AIC and BIC in this case and how could they go further ?*
**<Authors' reply>** The use of AIC,BIC criteria is guided by best practices in the domain of nonstationary extreme value analysis (e.g., Kim et al., 2017; Salas and Obeysekera, 2014), and more particularly recommended for choosing among various fragility models (e.g. Lallemant et al., 2015); see also an application of these criteria in the domain of nuclear safety by Zentner (2017). We agree however with referee #1 that further explanations should be given regarding model selection based on information criteria, because the perspectives differ when using AIC or BIC:

- On the one hand, AIC-based analysis considers a model to be a probabilistic attempt to approach the infinitely complex data-generating truth – but only approaching not representing (Höge et al. 2018: Table 2). This means this type of analysis aims at addressing which model will best predict the next sample, i.e. it provides a measure the predictive accuracy of the different models (Aho et al., 2014: Table 2);
- On the other hand, the purpose of BIC-based analysis considers each model as a probabilistic attempt to truly represent the infinitely complex data-generating truth assuing that the true model exists and is among the candidate models (Höge et al. 2018: Table 2). This perspective is different from the one of AIC and focuses in an approximation of the marginal probability of the data (here lEDP) given the model (Aho et al., 2014: Table 2); hence giving some insights to address which model generated the data, i.e. it measures goodness of fit.

Testing both criteria, AIC or BIC, thus provides both visions on the problem of model selection. This is now described in a new sub-sect. 2.2 entitled "Model Selection."

Regarding the comment on QQ plot, we agree that it can be used to validate model with the goal of explaining the observations. Yet, we only partly agree with referee #1 about its role for model selection. In current practices, QQ plots are rather used for model checking and not model selection, i.e. to control the model fit by examining the residuals once the model has been selected, and to identify why the model is adequate (or not). Its effectiveness has clearly been shown in statistical literature, but when restricted to visual inspection (see e.g. Loy et al., 2016). We propose to complement this diagnostic by the analysis of the PP plot as well. This will enable us to better emphasize the goodness of fit for large quantile levels. See below an example of presentation.

[Figure]

**Figure:** Diagnostic plots to check the validity of the considered model: (a) QQ plot for the deviance residuals for the NOsmo2 model; (b) QQ plot for the deviance residuals for the GEVsmo2 model with epistemic uncertainty; (c) QQ plot on Gumbel scale; (d) PP plot on Gumbel scale.

It should also be underlined that using QQ(or PP) plots only account for one part of the problem of model selection, i.e. goodness of fit. It does not account for the complexity of the considered model contrary to information criteria like BIC (or AIC). The advantage of information criteria is to include the first aspect (or predictive capability when using AIC) but also a correction related to the complexity of the model; here provided by the number of model parameters. Bayesian information criterion generally penalizes more complex models more strongly than does the AIC.

Finally, we now clearly underline the difficulty in some situations to perform the model selection when the AIC/BIC values are close in Sect. 2.2 as follows: "Yet, selecting the most appropriate model may not be straightforward in all situations when two model candidates present close AIC/BIC values. For instance, Burnham & Anderson (2004) suggests a AIC difference of at least 10 to be able to rank with confidence the most and the second most appropriate model. Otherwise, we propose complement the analysis by the likelihood ratio test LRT (e.g., Panagoulia et al., 2014: Sect. 2), which compares two hierarchically nested GEV formulations using $L=-2(l_0-l_1)$, where $l_0$ is the maximized log-likelihood of the simpler model M0 and $l_1$ is the one of the more complex model M1 (that presents q additional parameters compared to M0 and contains M0 as a particular case). The criterion L follows a chi-squared distribution with q degrees of freedom, which allows deriving a p-value of the test". Also note that we preferably analyse in Fig. 6 and 10 the AIC/BIC values relative to the minimum value.

Regarding the comment on the selection of the Gumbel model, we agree with referee #1. This aspect was outlined in the original version of the manuscript as follows: "The estimated shape parameter reaches here a constant value of 0.07 (+/-0.05), hence indicating a behaviour close to the Gumbel domain". This result could be seen as an additional element supporting the flexibility of the proposed approach based on GEV, which encompasses the Gumbel distribution as a particular case.

**References**

Aho, K., Derryberry, D., & Peterson, T. (2014). Model selection for ecologists: the worldviews of AIC and BIC. Ecology, 95(3), 631-636.

Höge, M., Wöhling, T., & Nowak, W. (2018). A primer for model selection: The decisive role of model complexity. Water Resources Research, 54(3), 1688-1715.

Kim, H., Kim, S., Shin, H., & Heo, J. H. (2017). Appropriate model selection methods for nonstationary generalized extreme value models. Journal of Hydrology, 547, 557-574.

Lallemant, D., Kiremidjian, A., & Burton, H. (2015). Statistical procedures for developing earthquake damage fragility curves. Earthquake Engineering & Structural Dynamics, 44(9), 1373-1389.

Loy, A., Follett, L., & Hofmann, H. (2016). Variations of Q–Q Plots: The power of our eyes!. The American Statistician, 70(2), 202-214.

Salas, J. D., & Obeysekera, J. (2014). Revisiting the concepts of return period and risk for nonstationary hydrologic extreme events. Journal of Hydrologic Engineering, 19(3), 554-568.

Zentner, I. (2017). A general framework for the estimation of analytical fragility functions based on multivariate probability distributions. Structural Safety, 64, 54-61.

**2.2 Application case**

*The selection of the ground-motion records if not described precisely enough from my point of view, for example the scaling levels are not stated. E.g. return levels for with quantity ? The records are non-linear and non-stationary in time, so how the spectrum is computed and scaled? The computational time for running seems to be omitted, it might be interesting to give an idea if a more important database could be generated.*

**<Authors' reply>** We thank the reviewer for his/her interest in the inner workings of the ground-motion selection procedure: we agree that the conditional spectrum approach used here is currently not sufficiently described. Therefore, we propose to replace the initial text in lines 212-222 by the following text:

"Thanks to the consideration of reference earthquake scenarios at various return periods, the scaling of a set of natural records is carried out to some extent, while preserving the consistency of the associated response spectra. The steps of this procedure hold as follows:

- *Choice of a conditioning period*: the spectral acceleration (SA) at $T^* = 0.38$s (fundamental mode of the structure) is selected as the ground-motion parameter upon which the records are conditioned and scaled.
- *Definition of seismic hazard levels*: six hazard levels are arbitrarily defined, and the associated annual probabilities of exceedance are quantified with the OpenQuake engine (www.globalquakemodel.org), using the SHARE seismic source catalogue (Woessner et al., 2013), for an arbitrary site in Southern Europe. The GMPE from Boore et al. (2014) is used to generate the ground motions, assuming soil conditions corresponding to $V_{s,30} = 800$ m/s at the considered site. Data associated with the mean hazard curve are summarized in Table 2.

*New Table 2: Estimation of the seismic hazard distribution for the application site.*

| Scaling level | SA(0.38s) [g] | Annual Probability of Exceedance | Return Period |
|---|---|---|---|
| #1 | 0.185 | 4.87E-2 | 20 y |
| #2 | 0.617 | 4.99E-3 | 200 y |
| #3 | 0.836 | 2.50E-3 | 400 y |
| #4 | 1.492 | 5.00E-4 | 2,000 y |
| #5 | 2.673 | 5.00E-5 | 20,000 y |
| #6 | 3.882 | 5.00E-6 | 200,000 y |

- *Disaggregation of the seismic sources and identification of the reference earthquakes*: the OpenQuake engine is used to perform a hazard disaggregation for each scaling level. A reference earthquake scenario may then be characterized through the variables [$M_w$; $R_{jb}$; $\varepsilon$] (i.e., magnitude, Joyner-Boore distance, error term of the ground-motion prediction equation), which are averaged from the disaggregation results (Bazzurro & Cornell, 1999). This disaggregation leads to the definition of a mean reference earthquake (MRE) for each scaling level.
- *Construction of the conditional spectra*: for each scaling level, the conditional mean spectrum is built by applying the GMPE to the identified MRE. For each period $T_i$, it is defined as follows (Lin et al., 2013):

$$\mu_{\ln SA(T_i)|\ln SA(T^*)} = \mu_{\ln SA}(M_w, R_{jb}, T_i) + \rho_{T_i, T^*} \cdot \varepsilon(T^*) \cdot \sigma_{\ln SA}(M_w, T_i) \qquad (10)$$

where $\mu_{lnSA}(M_w, R_{jb}, T_i)$ is the mean output of the GMPE for the MRE considered, $\rho_{T_i, T^*}$ is the cross-correlation coefficient between $SA(T_i)$ and $SA(T^*)$ (Baker & Jayaram, 2008), $\varepsilon(T^*)$ is the error term value at the target period $T^* = 0.38s$, and $\sigma_{lnSA}(M_w, T_i)$ is the standard deviation of the logarithm of $SA(T_i)$, as provided by the GMPE. The associated standard deviation is also evaluated, thanks to the following equation:

$$\boldsymbol{\mu_{ln\,SA(T_i)|ln\,SA(T^*)}} = \boldsymbol{\mu_{ln\,SA}\left(M_W, R_{jb}, T_i\right)} + \boldsymbol{\rho_{T_i, T^*} \cdot \varepsilon(T^*) \cdot \sigma_{ln\,SA}\left(M_W, T_i\right)} \tag{11}$$

The conditional mean spectrum and its associated standard deviation are finally assembled in order to construct the conditional spectrum at each scaling level. The conditional mean spectra may be compared with the uniform hazard spectra (UHS) that are estimated from the hazard curves at various periods. As stated in Lin et al. (2013), the SA value at the conditioning period corresponds to the UHS, which acts as an upper-bound envelope for the conditional mean spectrum.

- *Selection and scaling of the ground-motion records*: ground-motion records that are compatible with the target conditional response spectrum are selected, using the algorithm by Jayaram et al. (2011): the distribution of the selected ground-motion spectra, once scaled with respect to the conditioning period, has to fit the median and standard deviation of the conditional spectrum that is built from Eq. 7 and 8. The final selection from the PEER database (PEER, 2013) consists of 30 records for each of the 6 scaling levels (i.e., 180 ground-motion records in total).

The non-linear dynamic analyses are performed on a high performance-computing cluster, enabling the launch of the multiple runs in parallel (e.g., a ground-motion of a duration of 20s is processed in around 3 or 4 hours). Here, the main limit with respect to the number of ground-motion records is not necessarily related to the computation cost, but more to the availability of natural ground motions that are able to fit the conditional spectra at the desired return periods."

References:
Baker, J.W., & Jayaram, N. (2008). Correlation of spectral acceleration values from NGA ground motion models. Earthquake Spectra, 24(1), 299-317.
Bazzurro, P., & Cornell, C.A. (1999). Disaggregation of seismic hazard. Bulletin of the Seismological Society of America, 89(2), 501-520.
Jayaram, N., Lin, T., & Baker, J.W. (2011). A computationally efficient ground-motion selection algorithm for matching a target response spectrum mean and variance. Earthquake Spectra, 27(3), 797-815.
PEER (2013). PEER NGA-West2 Database, Pacific Earthquake Engineering Research Center, https://ngawest2.berkeley.edu.
Woessner, J., Danciu, L., Kaestli, P., & Monelli, D. (2013). Database of seismogenic zones, Mmax, earthquake activity rates, ground motion attenuation relations and associated logic trees. FP7 SHARE Deliverable Report D6.6.

**2.3 Results**

*The models compared here do not include parametric models (polynomials, nonlinear...) and the selected models are the non-linear smooth models. One question is related to the ability of this models to extrapolate beyond the range of variation of the training set? It might be interesting to compare to classical parametric models (if any) of with some polynomials models to also investigate the extrapolation ability.*

**<Authors' reply>** Regarding the problem of extrapolation ability, though of interest, we are not fully convinced that this is within the scope of our study.

- Considering the study without uncertainty, the selection of the ground-motion records is performed in order to cover a wide range of plausible earthquake scenarios (hence of PGA) in particular by considering an upper bound for PGA of 30 m/s² (i.e. ~3g) for return periods up to large values of 200,000 years (see reply to comment 2.2). Extrapolating outside this upper bound may not be considered physically realistic;

- Considering the study with uncertainty, the mechanical and geometrical parameters are here bounded and extrapolating outside the considered range may suffer from a lack of realism as well. In a more generic case, for instance with uncertainties represented by unbounded probability distributions, the problem could however appear. This aspect is now clearly outlined in the discussion section Sect. 5.

We however totally agree with referee #1 that the problem of extrapolation is more stringent when nonstationary is related to temporal covariates as outlined for instance by Salas and Obeysekera (2014).

Including a larger number of parametric models may be of interest, but the proposed models should remain realistic. Current practices in seismic vulnerability analysis mostly focus on simple linear models, because in most situations data suggest it and because they remain interpretable. Models of intermediate complexity like polynomial models of second order are rarely used. We choose not to include them in the current analysis.

In order to provide some elements to referee #1, the following figure shows that a second-order polynomial GEV model (denoted GEVpoly1,2 whether it is applied on $\mu$ or $\sigma$) is not identified by the AIC/BIC analysis as the "most appropriate" model (compared to the smooth GEV).

[Figure]

**Figure:** Model selection criteria (AIC (a) and BIC (b)) for the different models considering the derivation of a FC without parametric uncertainty.

In current practices, when non-linearity is suspected, more complex non-parametric models are generally preferred based for instance on neural networks (see e.g., Wang et al., 2018) or on kernel smoothing (see e.g. Mai et al., 2017) because they enable to derive from the data the non-linearity by avoiding to specify the form/shape of the non-linearity. This is also the advantage of the proposed approach. Comparison to these alternatives is here out of the scope of the present study and we choose to underline this perspective in the discussion section Sect. 5.

*If my understanding is correct, the uncertainties in the estimation of the marginal effects are neglected in computing the fragility curves, that is the reason why the are no uncertainties on figure 7. However, in figures 12 and 13, uncertainties linked to the variability of the input variables are shown. As is, it difficult to know which source of uncertainties is the highest and a discussion on this point would add a great value to the paper.*

- **<Authors' reply>** We totally agree with the referee #1 and have completed the analysis by incorporating the uncertainties on the regression coefficients (Eq. 5). The following procedure is conducted to account for the epistemic uncertainties:
  - Step 5.1: the considered *IM* is fixed at a given value;
  - Step 5.2: for the considered *IM* value, a large number (here chosen at $n$=1,000) of $U$ samples are randomly generated;
  - Step 5.3: for each of the randomly generated $U$ samples, the failure probability is estimated for the considered *IM* value;
  - Return to step 5.1.

The result of the procedure corresponds to a set of $n$ FCs from which we can derive the median FC as well as the uncertainty bands based on the pointwise confidence intervals at different levels. These uncertainty bands thus reflect the impact of the epistemic uncertainty related to the mechanical/geometrical parameters. Due to the limited number of observations, the derived FC is associated to the uncertainty on the fitting of the probabilistic model (e.g., GEV or Gaussian) as well. To integrate this fitting uncertainty in the analysis, step 5 can be extended by randomly generating parameters of the considered probabilistic model at step 5.2 (by assuming that they follow a multivariate Gaussian distribution).

Note that including the fitting error made us reconsider our coding, which enabled us to identify a bug in our original script. Contrary to the previous result, the differences GEV-based FC and the one based on the normal assumption are not so large as previously underlined.

In Sect. 4, we now discuss in more details two cases: (1) with uncertainty on the geometrical/mechanical parameters only (termed as "epistemic uncertainty"); (2) with the uncertainty on the fitting as well (i.e. uncertainty on the model parameters). See below a new presentation of the results (new Figure 14). This type of presentation is used to compare the implications of both types of uncertainty.

- For case #1 (without epistemic uncertainty), we point out in Sect. 4.1.3 (page 19, lines 405-420) that: "The resulting FC is compared to the one based on the normal assumption (Fig. 9b). This shows that $P_f$ would have been under-estimated for moderate-to-large PGA from 10 to ~25 m²/s if the selection of the probability model had not been applied (i.e. if the widespread assumption of normality had been used); for instance at PGA=20 m²/s, $P_f$ is under-estimated by ~5%. This is particularly noticeable for the range of PGA from 10 to 15 m²/s, where the GEV-based FC clearly indicates a non-zero probability value, whereas the Gaussian model indicates negligible probability values below 1%. For very high PGA, both FC models approximately provide almost the same $P_f$ value.

These conclusions should however be analysed with respect to the fitting uncertainty, which has here a clear impact; for instance at PGA=20 m²/s, the 90% confidence interval has a width of 10% (Fig. 9a), i.e. of the same order of magnitude than a PGA variation from 10 to 20 m²/s. We note also that the fitting uncertainty reaches the same magnitude between both models. This suggests that additional numerical simulation results are necessary to decrease this uncertainty source for both models";

- For case #2 (with epistemic uncertainty), we point out in Sect. 4.2.3 (page 25, lines 477-492) that: "We show that the GEV-based FC is less steep than the one for case #1 (Fig. 9): this is mainly related to the value of the shape parameter (close to Gumbel regime for case #1 without epistemic uncertainty and of Weibull regime for case #2 with epistemic uncertainty). Fig. 14a also outlines that the uncertainty related to the mechanical and geometrical parameters has a non-negligible influence as shown by the width of the uncertainty bands: for PGA=30 m²/s, the 90% confidence interval has a width of ~20%. In addition, the inclusion of the fitting uncertainty (Fig. 14b) increases the width of the confidence interval, but it appears to mainly impact the 90% confidence interval (compare the dark and the light coloured envelope in Fig. 14); for instance, compared to Fig. 14a, this uncertainty implies a +5% (respectively -5%) shift of the upper bound (respectively lower bound) of the 90% confidence interval at PGA=30 m²/s. Compared to the widely-used assumption of normality, Fig. 14c,d show that the failure probability reached with this model is larger than with the GEV-based FC; at PGA=30 m²/s, the difference reaches ~5%. In practice, this means that a design based on the Gaussian model would have here been too conservative. Regarding the impact of the different sources of uncertainty, the epistemic uncertainty appears to influence less the Gaussian model than the GEV one. The impact of the fitting uncertainty is however quasi-equivalent for both models".

Given the importance of the fitting uncertainty, we also identify in the discussion section some lines of improvement as follows (Sect. 5: page 28, lines 544-549): "The latter [variance-based sensitivity analysis] approach opens promising perspectives to improve the interpretability of the procedure by clarifying the respective role of the different sources of uncertainty, i.e. related to the mechanical/geometrical parameters, but also to the fitting process, which appears to have a non-negligible impact in our study. The modelling of this type of uncertainty is based here on the Gaussian assumption for the fitting errors. Bayesian techniques within framework of GAMLSS (Umlauf et al., 2018) is another line of improvement worth investigating."

[Figure]

**New Figure 14.** Fragility curve (relating the failure probability $P_f$ to *PGA*) considering epistemic uncertainties only (left), and fitting uncertainty as well (right). (a,b) GEV-based FC; (c,d) FC based on the normal assumption. The coloured bands are defined based on the pointwise confidence intervals derived from the set of FCs (see text for details).

**3. Technical remarks**
*- Both formula, figures and tables should be centered to be easier to read; - in figures 10 and 11,*

**\<Authors' reply\>** We apologize for this lack of readability but we followed the instructions provided in the word template of NHESS where the equations, tables and figures are formatted as "justified" (https://www.natural-hazards-and-earth-system-sciences.net/for_authors/manuscript_preparation.html).

*- some variables seems to be evenly distributed and some other (e.g. E_IC, nXi_RC, e5 in figure 10) seems to be random : it seems that all of them should be uniform of the range of variation stated in Table 1 ?*

**\<Authors' reply\>** We checked this aspect (see figure below) and it seems that it is only a visual effect related to the parametrisation of the ticks on Figure 10.

- *The link functions are not stated precisely in table 2;*
**<Authors' reply>** This is now specified.

**Referee #2:**

**1. General comments**

The paper proposes a method to derive improved and accurate fragility curves for nuclear plants subject to seismic activity by adopting non-stationary GEV for the engineering demand parameter. The capacity of the structure is simulated allowing for parametric uncertainties. The premise of the proposed method is novel and reasonable. The analysis is rigorous and encompasses major requirements of uncertainty quantification. The results indicate that the usage of the method presented may provide better vulnerability assessment in nuclear plant safety analysis by a significant margin. The paper may benefit from improved clarity of presentation, particularly in the final computation of fragility curves (FC) and presentation of results.

"**<Authors' reply>**" we are grateful to referee #2 for his/her analysis. The following document provides details on how we improve the different aspects, which consists in:
   - Improving the quality of the figures;
   - Adding a technical appendix on the double-penalisation procedure as well as a synthetic case to illustrate its effectiveness;
   - Adding details on the dynamic simulations and on their analysis to derive the fragility curves;
   - Clarifying the added value of our approach with respect to the literature.

**2. Specific comments: Methods**

The choice of whether to use GEV is done using AIC or BIC measures. The benefit to using a non-stationary GEV may be demonstrated by showing improved goodness of fit or other custom measures as applicable. Fig.5 and 9 may be locations to include such a comparison.

"**<Authors' reply>**" We agree with referee #2 that we should better emphasize the benefit of a non-stationary formulation.

To do so, we made the following modifications:

   - We included the stationary Normal and GEV model in the list of tested models (see new Table 3) and present now the results in terms of AIC/BIC differences relative to the minimum value (Fig. 6 and 10);

   - We complemented the procedure described in Sect. 2.1 as follows: "Yet, selecting the most appropriate model may not be straightforward in all situations when two model candidates present close AIC/BIC values. For instance, Burnham & Anderson (2004) suggests a AIC difference of at least 10 to be able to rank with confidence the most and the second most appropriate model. Otherwise, we propose complement the analysis by the likelihood ratio test LRT (e.g., Panagoulia et al., 2014: Sect. 2), which compares two hierarchically nested GEV formulations using $L=-2(l_0-l_1)$, where $l_0$ is the maximized log-likelihood of the simpler model M0 and $l_1$ is the one of the more complex model M1 (that presents q additional parameters compared to M0 and contains M0 as a particular case). The criterion L follows a chi-squared distribution with q degrees of freedom, which allows deriving a p-value of the test".

The LRT test was applied on case #1 given the small AIC/BIC differences for models GEVsmo2 and GEVsmo3. For illustration, we then outline in Sect. 4.1.1 that "the LRT p-value for a stationary GEV model and the non-stationary GEVsmot2 model is here far less than 1%".

**References**
Panagoulia, D., Economou, P., & Caroni, C. (2014). Stationary and nonstationary generalized extreme value modelling of extreme precipitation over a mountainous area under climate change. Environmetrics, 25(1), 29-43.

The derivation of the fragility after arriving at demand and capacity may be done more explicitly. The nonlinear structural analysis section describes a scaling range with 6 steps. One might expect a few data points on the FC rather than a continuous curve based on this. A fit may be done after this and superimposed on the same graph. The absence of this plot may be due to the usage of a different method. The GEV fit appears to be for the EDP but is also mentioned as the fit for FC. Clarity here may improve readability considerably.

**<Authors' reply>** We thank referee #2 for his/her on-point comment regarding FC derivation methods. What the reviewer describes here pertains to the derivation of FC using multi-stripe analysis or incremental dynamic analysis (Baker, 2015; Vamvatsikos & Cornell, 2002): multiple ground-motion are scaled at the same IM value, and statistic on the exceedance rate of a given EDP value may be extracted at each IM step. In our present study, the conditional spectrum method leads to the selection and scaling of ground motions with respect to SA(0.38s), which correspond to the fundamental modal of the structure. However, the fragility analysis is focused on the pipeline component (located along the structure), which appears to be more susceptible to PGA: therefore, PGA is chosen as IM in the present fragility analysis. For this reason, it is not feasible to represent probabilities at 6 levels of PGA (as shown in Figure 3, there is a variability around the 6 scaling levels). In this case, current approach for FC derivation are the 'regression on the IM-EDP cloud' (i.e., least-squares regression, as demonstrated by Cornell et al., 2002) or the use of GLM regression or maximum likelihood estimation (Shinozuka et al., 2000). We now add a new figure (Fig. 4) showing the probabilities at the 6 selected return periods (which may be associated to unique values of SA(0.38s), but not of PGA).

[Figure]

**New Figure 4.** (a) Damage probabilities directly extracted from the 6 scaling levels (or return periods); (b) Damage probabilities w.r.t. the 6 SA(T*) levels, and fitted lognormal cumulative distribution function.

From Figure 4, two main observations can be made: *(i)* the multiple stripe analysis does not emphasize any different between the models with and without parametric uncertainty, and *(ii)* the FC directly derived from the 6 probabilities does not provide a satisfying fit. This is now outlined in Sect. 3.1.

**References**
Baker, J. W. (2015). Efficient analytical fragility function fitting using dynamic structural analysis. Earthquake Spectra, 31(1), 579-599.
Cornell, C.A., Jalayer, F., Hamburger, R.O., & Foutch, D.A. (2002). Probabilistic basis for 2000 SAC federal emergency management agency steel moment frame guidelines. Journal of Structural Engineering, 128(4), 526-533.
Shinozuka, M., Feng, M., Lee, J., & Naganuma, T. (2000). Statistical analysis of fragility curves. Journal of Engineering Mechanics, 126(12), 1224-1231.
Vamvatsikos, D., & Cornell, C.A. (2002). Incremental dynamic analysis. Earthquake Engineering & Structural Dynamics, 31(3), 491-514.

The same ambiguity arises in the plots of partial effects (Figs. 6,10,11). The structural variables appear to have a partial effect on the demand parameter. This seems counter-intuitive conventionally. The GEV appears to be used not just for demand in that case, this may be better presented.

"**<Authors' reply>**" We agree with referee #2 that there is some ambiguity on that aspect. Similarly as for the previous comment, we propose to better clarify in the introduction (page 1-2, lines 30-35) that we focus on the derivation of an appropriate statistical model for demand parameter based on which we derive the FC. The introduction is now completed by referring to alternatives like incremental dynamic analysis (page 3, lines 77-84) (when describing limit (4) in the introduction).

The convolution of the probability density of capacity around the pre-defined damage states and the 1-CDF of the demand on the structure by different levels of ground motion would produce points on the FC. The procedure is detailed in [1]. This may be used as a starting point to show how FC derivation is different here.

"**<Authors' reply>**" Thank you for this valuable reference. As far as we understand the suggested study, Rota and co-authors account for the uncertainty on the structure by deriving the statistical law of the different damage states (Fig. 11 of their paper). In our study, we considered a fixed threshold *th* for defining the damage state (here fixed at *th*=775 kN). Combining the proposed approach with the one of Rota and co-authors is clearly worth investigating. It should however be underlined that the translation is not direct, because we actually performed dynamic numerical simulations, whereas Rota and co-authors based their procedure on pushover analyses. Analysing how to make this link may deserve further developments. We propose to add a reference to this approach in the introduction and to point out this aspect in the discussion section (Sect. 5, 529-534, pages 27-28) as follows: "As indicated in the introduction, an alternative approach would rely on the principles of the IDA method; the advantage being to capture the variability of the structural capacity and to get deeper insight into the structural behaviour. See an example for masonry buildings by Rota et al. (2010). Yet, the adaptation of this technique would impose additional developments to properly characterise collapse through the numerical model (see discussion by Zentner et al.,

2017: Sect. 2.5). Sect. 3.1 also points out the difficulty in applying this approach in our case. Combining the idea underlying IDA and our statistical procedure is worth investigating in the future".

Return period for a non-stationary model requires transformation which may be of significance in some cases. See [2]. This may be of no effect considering the order of scaling, but it may be of use to include/discuss.

"**<Authors' reply>**" Thank you for this suggestion and reference, which was included in the discussion section as a possible option to enhance the interpretability of the results in terms of covariates' influence on FC. This is done on page 28, lines 540-544 as follows "analysing the role of each covariate from a physical viewpoint, as done for instance by Salas and Obeysekera (2014) to investigate the evolution of hydrological extremes over time (e.g. increasing, decreasing or abrupt shifts of hydrologic extremes). Some valuable lessons can also be drawn from this domain of application to define and communicate an evolving probability of failure."

**3. Readability**

The paper may benefit from an appendix or a different section for detailed methods after describing the main results.

"**<Authors' reply>**" An appendix has been added to give further details on the double penalisation procedure.

The partial effects may be consolidated in one section, the fragility curves being in another.

"**<Authors' reply>**" Thank you for this suggestion. Sect. 4.X (X=1,2) are now subdivided in as follows: Sect. 4.X.1: Model selection; Sect. 4.X.2: Partial effects; Sect. 4.X.3: FC derivation.

The variation of the fragility curves based on the choice for parameters such as e4 may be better presented in measures of percentage changes.

"**<Authors' reply>**" Thank you for this suggestion, which has been taken into account. New Figure 15 should be presented as follows.

[Figure]

**New Figure 15.** FC considering different thickness e4: (a) -12.5% of the original value; (b) -5%; (c) +5%; (d) +12.5%. Uncertainty bands are provided by accounting for epistemic uncertainty only (dark blue) and by accounting for the fitting uncertainty as well (light blue).

The method used by Wood et al. (line 97) could've included with more detail for completeness.

"**<Authors' reply>**" On the one hand, adding details (in a new appendix) on the double penalisation approach is clearly necessary, because it is a key ingredient of the proposed procedure. On the other hand, adding too many details on developments (that we did not perform), might hamper the readability of the paper. This may be the case for the AIC

formulation for GAM models; the interested reader can easily find such details in the indicated reference.

The paper may benefit from a tabular presentation of results, especially the effect of structural parameters on FC as this may be of key significance for a practitioner.
"**<Authors' reply>**" Thank you for this suggestion. We now present the effect of the structural parameters on FC using the following new Table 4.

New Table 4. Influence of the geometrical/mechanical parameters on the GEV parameters, $\mu$ and $l\sigma$ of the GEVsmo2 model

| Variable | Influence on $\mu$ | Influence on $l\sigma$ |
|---|---|---|
| $E_{IC}$ | Linear (decreasing) | - |
| $\xi_{RPC}$ | - | Non-linear (non-monotone) |
| $\xi_{RC}$ | Non-linear (non-monotone) | Non-linear (decreasing) |
| $e_1$ | Linear (increasing) | - |
| $e_2$ | - | - |
| $e_3$ | Linear (decreasing) | - |
| $e_4$ | Linear (increasing) | - |
| $e_5$ | Linear (increasing) | Non-linear (non-monotone) |
| $e_6$ | Non-linear (non-monotone) | - |
| $\xi_{SL}$ | - | - |

Figure quality may be improved. Consider using vector graphics. x-axis of Fig. 5 requires uniformity and units may be placed in brackets.
"**<Authors' reply>**" Thank you for these suggestions that will be take into account. The image are embedded as *png* format in the text for sake of file size and vector graphics are provided as separate files.

[1] Rota, M., A. Penna, and G. Magenes. "A methodology for deriving analytical fragility curves for masonry buildings based on stochastic nonlinear analyses." Engineering Structures 32.5 (2010): 1312-1323.

[2] Salas, Jose D., and Jayantha Obeysekera. "Revisiting the concepts of return period and risk for nonstationary hydrologic extreme events." Journal of Hydrologic Engineering 19.3 (2014): 554-568.

March 23[rd], 2020

Jeremy Rohmer[1], Pierre Gehl[1], Marine Marcilhac-Fradin[2], Yves Guigueno[2], Nadia Rahni[2], Julien Clément[2]

[1]BRGM, 3 av. C. Guillemin, 45060 Orléans Cedex 2, France
[2]Institute for Radiological Protection and Nuclear Safety, Fontenay-Aux-Roses, 92262, France

---

## Author Response (AR2)

**Replies to the reviewers' comments on "Non-stationary extreme value analysis applied to seismic fragility assessment for nuclear safety analysis". (nhess-2019-400)**

We would like to thank referee #3 for his/her constructive comments. We agree with most of the suggestions and, therefore, we have modified the manuscript to take on board their comments (outlined in green). In the following, we recall the reviews and we reply to each of the comments in turn (outlined by "**<Authors' reply>**").

**Please note that the line numbers of changes are indicated and correspond to the revised manuscript with marked changes.**

**Referee #3:**

In this paper, the authors investigate how the tools of non-stationary extreme value analysis can be used to model in a flexible manner the tail behaviour of the engineering demand parameter as a function of the considered intensity measure. The focus of the analysis is the dynamic response of an anchored steam line and of a supporting structure under seismic solicitations.

I recommend the publication of this paper after minor revisions

Comments:

*1. The paper describes very carefully all the assumptions and drawbacks of the proposed method, compared to the traditional strategy to compute Fragility Curves.*
*I believe that section 3 can be improved by supporting the choice of the mechanical and geometrical parameters of interest for the quantification of the epistemic uncertainty, with some sensitivity analysis (Sobol's indices for instance). In this way, the influence of each parameter can be assessed and the whole problem dimensionality (maybe) reduced before hand (as the authors recognized in section 4.2). Is there any correlation between EDP and the key mechanical and geometrical parameters?*
**<Authors' reply>** Thank you for the suggestion that we find very valuable. We totally agree that a global sensitivity analysis would be useful to identify beforehand some key mechanical/geometrical parameters. This is now clearly outlined in the discussion section 5, page 28, lines 544-546 as follows "The latter approach [variance-based global sensitivity analysis] opens promising perspectives to ease the fitting process by filtering out beforehand some negligible mechanical/geometrical parameters. It is also expected to improve the interpretability of the procedure by clarifying the respective role of the different sources of uncertainty i.e. related to the mechanical/geometrical parameters, but also to the fitting process, which appears to have a non-negligible impact in our study".

Using linear correlation may be a solution to achieve this objective, but unfortunately, in our case, the matrix of Pearson correlation coefficient clearly indicates the lack of such linear relation. See figure below where the color indicates the magnitude of the coefficient, and the cross-like marker indicates where the coefficient is insignificant (at 1% level).

[Figure]

*3. It would be very interesting to see whether synthetic input ground motion time-histories can improve the database concistency, especially for larger values of PGA, where the EDP dispersion seems larger.*

**<Authors' reply>** We agree that using synthetic input ground motion time histories could be an alternative approach. It has been decided however to use only natural records in the present application, in order to accurately represent the inherent variability of other ground motion parameters such as duration. This is now clearly indicated in Sect. 3.2, page 11, lines274-276.

We also added two references on those aspects, namely:

Pousse, G., Bonilla, L.F., Cotton, F., Margerin L.: Nonstationary stochastic simulation of strong ground motion time histories including natural variability: application to the K-net Japanese database, Bull. Seismol. Soc. Am., 96, 2103–2117, 2006.

Boore, D.M.: Simulation of ground motion using the stochastic method, Pure and applied geophysics, 160(3-4), 635-676, 2003.

*4. I suspect that the large influence of the damping coefficient on the GEV model might hide some non-linear effects taking place and/or some simplified assumption in the coupling method between structural dynamic response and anchored steam line. Could you clarify on this?*

**<Authors' reply>** Thank you for this comment and for suggesting the possible explanation. We however recognize that the interpretation is difficult to conduct and further investigations are here necessary. At least, this exemplifies one advantage of the proposed procedure; this problem could not have been identified if the analysis of the partial effect had not been done.

We clarify this aspect as follows (Sect. 4.2.1, page 23-24, lines 465-475) "We show here that a larger number of input parameters were filtered out by the selection procedure i.e. only the thickness $e_5$ is selected as well as the damping ratios of the concrete structures $\xi_{RPC}$ and $\xi_{RC}$ (related to the containment building). The partial effects are all non-linear, but with larger

uncertainty than for the location parameter (compare the widths of the red-coloured uncertain bands in Fig. 12 and 13). In particular, the strong non-linear influence of $\xi_{RPC}$ and $\xi_{RC}$ may be due to the simplified coupling assumption between structural dynamic response and anchored steam line (i.e., the displacement time-history at various points of the building is directly used as input for the response of the steam line). Identifying this problem is possible thanks to the analysis of the partial effects, though it should be recognized that this behavior remains difficult to interpret and further investigations are here necessary".

*4. Finally, I believe that the final discussion should provide some hints on the use of surrogate/multi-fidelity modelling to fasten the sampling task and increase the number of realizations instead.*
We agree with this suggestion. We added in the discussion section the following aspects (page 28, 549-554): "The treatment of this type of uncertainty can be improved on two aspects: 1) it is expected to decrease by fitting the FC with a larger number numerical simulation results. To relieve the computational burden (each numerical simulation has a computation time cost of several hours, see Sect. 3.2), replacing the mechanical simulator by surrogate models (like neural network, Wang et al., 2018 or using model order reduction strategy, Bamer et al., 2017) can be envisaged; 2) the modelling of such uncertainty can be done in a more flexible and realistic manner (compared to the Gaussian assumption made here) using Bayesian techniques within framework of GAMLSS (Umlauf et al., 2018)".

**Added reference:**
Bamer, F., Amiri, A. K., and Bucher, C.: A new model order reduction strategy adapted to nonlinear problems in earthquake engineering, Earthquake engineering & structural dynamics, 46(4), 537-559, 2017.

April 3[th], 2020

Jeremy Rohmer[1], Pierre Gehl[1], Marine Marcilhac-Fradin[2], Yves Guigueno[2], Nadia Rahni[2], Julien Clément[2]

[1]BRGM, 3 av. C. Guillemin, 45060 Orléans Cedex 2, France
[2]Institute for Radiological Protection and Nuclear Safety, Fontenay-Aux-Roses, 92262, France